

# Experimental and model-based investigation of the links between snow bidirectional reflectance and snow microstructure

Marie Dumont [1], Frederic Flin [1], Aleksey Malinka [2], Olivier Brissaud [3], Pascal Hagenmuller [1], Philippe Lapalus [1], Bernard Lesaffre [1], Anne Dufour [1], Neige Calonne [1], Sabine Rolland du Roscoat [4], and Edward Ando [4]

[1]Univ. Grenoble Alpes, Université de Toulouse, Météo-France, CNRS, CNRM, Centre d'Études de la Neige, 38000 Grenoble, France
[2]Institute of Physics, National Academy of Sciences of Belarus, Minsk, Belarus
[3]Univ. Grenoble Alpes - CNRS, IPAG, Grenoble, France
[4]UGA - Grenoble INP - CNRS, 3SR UMR 5521, Grenoble, France

*Correspondence to:* marie dumont (marie.dumont@meteo.fr)

**Abstract.** Snow stands out from materials at the Earth's surface owing to its unique optical properties. Snow optical properties are sensitive to the snow microstructure, triggering potent climate feedbacks. The impacts of snow microstructure on its optical properties such as reflectance are, to date, only partially understood. However, precise modelling of snow reflectance, particularly bidirectional one, are required in many problems, e.g. to process correctly satellite data over snow-covered areas. This study presents a dataset that combines bidirectional reflectance measurements over 500-2500 nm and the X-ray tomography of the snow microstructure for three snow samples of two different morphological types. The dataset is used to evaluate the stereological approach from Malinka (2014) that relates snow optical properties to the chord length distribution in the snow microstructure. The mean chord length and SSA retrieved with this approach from the albedo spectrum and those measured by the X-ray tomography are in excellent agreement. The analysis of the 3D images has shown that the random chords of the ice phase obey the gamma distribution with the shape parameter $m$ taking the value approximately equal or a little greater than 2. For weak and intermediate absorption (high and medium albedo), the simulated bidirectional reflectances reproduce the measured ones accurately but tend to slightly overestimate the anisotropy of the radiation. For such absorptions the use of the exponential law for the ice chord length distribution instead of the one measured with the X-ray tomography does not affect the simulated reflectance. In contrast, under high absorption (albedo of a few percent), snow microstructure and especially facet orientation at the surface, plays a significant role for the reflectance, particularly at oblique viewing and incidence.

## 1 Introduction

Snow optical properties are crucial to quantify the effect of snow cover on the Earth energy balance. They are also unique since snow is the most reflective material on the Earth surface. The subtle interplays between snow microstructure and snow optical properties are responsible for several climate feedbacks (e.g. Flanner et al., 2012). The dependencies of snow reflectance on snow microstructure have been already widely studied. It has been shown that the snow reflectance in the visible and near infrared region are primarily determined by the effective grain size that is defined as the ratio of the particle volume to its





average projected area (Grenfell and Warren, 1999; Kokhanovsky and Zege, 2004) and is uniquely related to the specific surface area, hereafter SSA, defined as the ratio between the ice/air interface area and the mass of a snow sample (e.g. Flin et al., 2004; Domine et al., 2006; Gallet et al., 2009).

The effect of snow microstructure on the optical properties of snow is currently not fully understood. Up to now, many

studies have focused on retrieving the single scattering properties of individual ice crystals with "idealised" shapes (e.g. Xie et al., 2006, Picard et al., 2009, Liou et al., 2011, Räisänen et al., 2015, Dang et al., 2016) and on using these calculations to infer the effect of crystal shapes on snow optical properties. Several studies have already shown that the effect of shape is more pronounced on bidirectional reflectance (e.g. Dumont et al., 2010) and the vertical irradiance profile in the snowpack (e.g. Libois et al., 2013) than on hemispherical reflectance (albedo). Alternative approaches include running ray-tracing models

directly on 3D images of the snow microstructure as done by Kaempfer et al. (2007) or at the intermediate level as done by Haussener et al. (2012) and Varsa et al. (2021). More recently, Ishimoto et al. (2018) used X-ray tomography images of different snow types and a ray tracing model to compute the single-scattering properties of snow particles. They found that the modeled orientation averaged scattering phase functions at two wavelengths (532 nm and 1242 nm) exhibit only a slight variation with the particle shapes.

Understanding and modelling the variations of snow directional reflectance with snow microstructure is essential to correctly interpret satellite data (Schaepman-Strub et al., 2006). Moreover, the sensitivity of snow directional reflectance to crystal shapes at least for high absorptive wavelengths (Xie et al., 2006; Dumont et al., 2010; Krol and Löwe, 2016a) makes snow directional reflectance a good candidate to provide an objective measurement of snow morphology. This characterisation is often performed using the grain types defined in Fierz et al. (2009). Such an approach however depends on the observer and

does not provide a variable that can be measured objectively and used directly in snowpack detailed modelling. Stanton et al. (2016) investigated the relationship between the BRF and various snow crystal morphologies based on measurements in the 400-1300 nm range. They concluded that, as surface hoar grows, the snow surface becomes less and less Lambertian but that the geometry (illumination and viewing), at which the reflectance is maximum or minimum is difficult to predict. Horton and Jamieson (2017) used reflectance measurements and investigated the potentiality of normalised difference indices calculated

from conical reflectance measurements at two wavelengths (860 and 1310 nm) to classify different crystal morphologies. They concluded that the bidirectional reflectance properties for different snow types must be investigated further.

The formalism developed by Kokhanovsky and Zege (2004) provides an analytical formulation linking the snow single scattering properties and reflectance to the effective grain size and a shape parameter $B$, a.k.a. absorption enhancement parameter. This formulation has been used in the snow radiative scheme TARTES (Libois et al., 2013) enabling to retrieve $B$ values from

concomitant measurements of irradiance profiles and reflectances on "homogeneous" snow layers. Though the obtained $B$ values (from 1.4 to 1.8) significantly differ from the $B$ value for spheres (1.25), no clear relationship has been established between grain types as defined in Fierz et al. (2009) and $B$ (Libois et al., 2014b). It must be underlined that this approach applies to low absorptions only, as $B$ is the proportionality factor in the first (linear) term of the power expansion of the absorption coefficient of snow in the absorption coefficient of ice.





More recently, Malinka (2014) used the stochastic approach that considers a porous material as a random two-phase mixture and directly relates its optical properties to the chord length distribution (CLD) in the medium. This approach is not restricted to low absorptive wavelengths and directly relates the snow optical properties to the snow microstructure by means of the CLD, e.g. the shape parameter $B$ for the random mixture arises in a natural way and equals $n^2$, where $n$ is the refractive index of ice. For the random mixture with $n = 1.31$, $B = 1.72$, while measurements in real snow performed both in a laboratory and *in situ* gave values of B ranged from 1.4 to 1.8 (Libois et al., 2014a). The approach has been successfully evaluated with respect to reflectance measurements over sea ice by Malinka et al. (2016).

Later on, Krol and Löwe (2016a) used X-ray tomography images to compare different metrics of snow microstructure. They experimentally demonstrated that the second moment of the CLD, $\mu_2$, can be related to a curvature length and also theoretically to the absorption enhancement parameter $B$. They theoretically predicted, using Malinka (2014)'s framework, that this microstructure metric strongly influences snow optical properties for high absorptive wavelengths. They also showed that the deviation of the CLD from the exponential law, which can be calculated using $\mu_2$, varies with snow types. However, no measurement of snow optical properties was used in this study to evaluate the validity of their findings.

To sum up, it has been shown that BRF, especially in high absorptive wavelengths, is more sensitive to snow morphology than snow albedo (bi-hemispherical reflectance). Yet no clear relationship has been either established theoretically or evaluated experimentally using optical measurements combined with an objective quantification of the snow microstructure. The objectives of the paper are thus to (i) describe one of the very few datasets that combined measurements of the bidirectional reflectance over 500-2500 nm range and X-tomography characterisation of the snow microstructure, (ii) evaluate the accuracy of Malinka (2014)'s model to simulate the snow BRF and its dependencies on snow microstructure parameters, and (iii) investigate the relationship between bidirectional reflectance and the snow microstructure beyond SSA using both simulation and measurement.

The first section provides a description of the snow samples, the measurements strategy, the optical model, the processing of the X-ray tomography images, and the optical data. The second section presents the results in terms of temporal variability of the snow microstructure, accuracy of the optical measurements, snow microstructure characterisation, and model evaluation. Discussions and conclusions are detailed in the last section.

## 2 Materials and methods

### 2.1 Experimental set-up and samples description

The general idea of the experiment is to characterize both the snow microstructure and the BRF for the same macroscopic snow sample. The dataset consists of 3 macroscopic snow samples: S1 analyzed in March 2012 and S2 and S3, the upper and lower surface of the same snow layer, analyzed in March 2013. S1 has been taken on natural new snow and can be classified as decomposing and fragmented particles/rounded grains (DF/RG) according to the classification of Fierz et al. (2009). S2 consisted of faceted crystals/depth hoar (FC/DH) obtained in a temperature gradient experiment. S3 is taken from the same



temperature gradient experiment as S2 except that it was turned upside-down so that the grain orientation is changed by 180°. For each sample, the experiment consists of the following steps (see subsections 2.1.1 to 2.1.3 for more detail):

1. Snow sample preparation.

2. Snow microstructure characterisation (manual measurements, casting for X-ray analysis) and preparation of a sample for BRF measurements.

3. BRF measurements.

4. Snow microstructure characterisation (manual measurements, casting for X-ray analysis).

Steps 2 and 4 were performed to characterise the microstructure just before and after the BRF measurements, and to control the possible evolution of the microstructure during the BRF measurements.

### 2.1.1 Snow samples preparation

Figure 1 provides pictures of the snow from each sample. For S1, a 7 cm thick snow layer was collected on a $60 \times 60$ cm$^2$ styrodur plate after a snowfall close to the lab and stored for 3 weeks in isothermal conditions at -20° C (Fig. 1A). For S2 and S3 (Fig. 1B), fresh snow was collected in the field and was sieved into the temperature gradient box from Calonne et al. (2014a) ($105 \times 58 \times 17.5$ cm$^3$). A vertical temperature gradient of $\approx 19.4$ C°m$^{-1}$ was applied inside the box with a mean temperature of -4° C. S2 and S3 had been stored under the described conditions for 16 and 18 days, respectively, before the measurements were made.

### 2.1.2 Snow sampling and manual characterisation

For the BRF measurements, the snow was sampled in a cylindrical sampler with no disturbance of the snow surface as in Dumont et al. (2010) (Fig. 1B). The diameter of the cylinder was $28 \pm 1$ cm. For S1 the snow thickness was 6.5 cm and for S2 and S3 it was 16.5 cm.

Before this sampling, the snow SSA and density were measured, the SSA with DUFISSS (DUal Frequency Integrating Sphere for Snow SSA measurement, Gallet et al., 2009) and ASSSAP (Alpine/Arctic Snow Specific Surface Area, Arnaud et al., 2011) and the density with manual weighing. Some small near-surface samples were taken for the X-ray analysis and casted using chloronaphthalene (chl) to block snow metamorphism (Flin et al., 2003; Calonne et al., 2014b). These measurements and sampling were reconducted after the optical measurements using the snow sampled in the big cylinder sampler.

### 2.1.3 X-ray tomography and BRF measurements

For all the samples, the X-ray tomography was performed at 7 µm resolution. Additionally, higher and lower resolutions (5 µm and 10 µm) were acquired for S1 and S2. The scanned samples were composed of three materials, namely ice, chloronaphthalene (chl), and some residual air bubbles due to incomplete impregnation of the sample (Flin et al., 2003; Hagenmuller et al.,



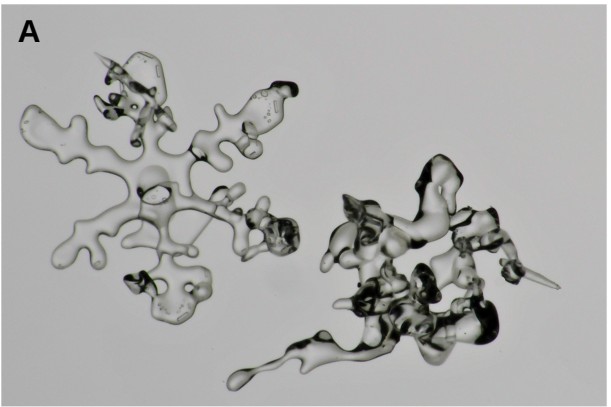

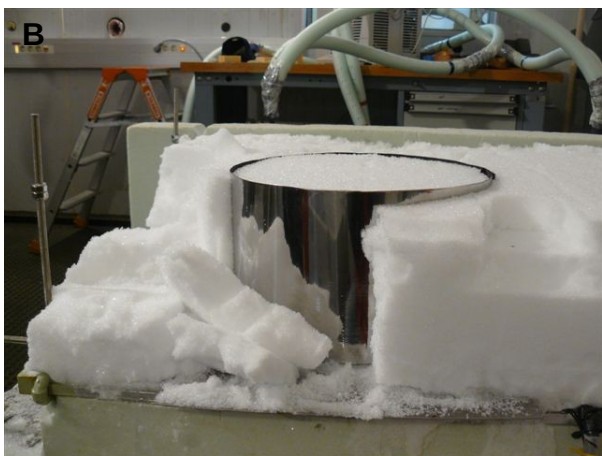

**Figure 1.** (A) Picture of snow from S1 taken with a microscope. (B) Experimental set-up for S2 sampling. The picture shows the inner part of the temperature gradient box (Calonne et al., 2014a) and the metallic sample holder for the optical measurements.

2013). These three materials can be distinguished by their X-ray attenuation coefficient, i.e. their grayscale value I. Table 1 provides an overview of all the images taken for the 3 samples and Figure 2 shows a sub-sample of the 3D images obtained for each sample. The image name provides the sample name, the timing of the scan (B for "before the start of the BRF measurements", A for "after the end of the BRF measurements"), the location of the sampling (1 in the centre of the optical sample and 2 on the border of it) and the resolution in microns.

The bidirectional reflectance was measured with a sensor field of view of 2.05° using a set-up described in Dumont et al. (2010) and Brissaud et al. (2004) in a cold room at –10°C.

Tables A1, A2 and A3 in Appendix A give an overview of the characteristics of the optical measurements for the 3 samples. The total duration of the optical measurements is 41 h for S1, 45 h for S2 and 94 h for S3.





**Table 1.** Summary of all the X-ray tomography images acquired. 'B' and 'A' refer to 'before' and 'after' the optical measurement, respectively.

| Image name | Optical sample | Resolution | Date of sampling | Time of sampling |
|---|---|---|---|---|
| Units | (-) | (μm) | (-) | (-) |
| S1_B_1_7m | S1, before | 7.25 | 03/27 | 13:00 |
| S1_B_1_10m | S1, before | 9.71 | 03/27 | 13:00 |
| S1_A_1_10m | S1, after | 9.71 | 03/29 | 10:00 |
| S1_A_1_7m | S1, after | 7.25 | 03/29 | 10:00 |
| S1_A_2_7m | S1, after | 7.30 | 03/29 | 10:00 |
| S2_B_1_7m | S2, before | 7.06 | 03/20 | 15:00 |
| S2_B_2_7m | S2, before | 7.03 | 03/20 | 15:00 |
| S2_A_1_7m | S2, after | 7.07 | 03/22 | 16:00 |
| S2_A_1_5m | S2, after | 5.83 | 03/22 | 16:00 |
| S2_A_1_10m | S2, after | 11.65 | 03/22 | 16:00 |
| S3_B_1_7m | S3, before | 7.06 | 03/22 | 13:00 |
| S3_B_2_7m | S3, before | 7.06 | 03/22 | 13:00 |
| S3_A_1_7m | S3, after | 7.07 | 03/26 | 14:00 |

## 2.2 X-ray tomography: image processing and analysis

### 2.2.1 Image processing

All grey level images were segmented using the following 3-step semi-automatic method: (i) pre-processing of the image, including basic beam hardening and ring artefact corrections; (ii) detection of air bubbles and replacement of their levels by the mean grey value of 1-chloronapthalene (see Flin et al., 2003, 2004 and "METHOD/Threshold-based segmentation/Air bubble detection" in Hagenmuller et al., 2013); (iii) application of the Energy-based binary segmentation method of Hagenmuller et al. (2013), with a resolution parameter $r = 1.2$ voxel.

Once segmented, the obtained binary images can be described in terms of $I(\mathbf{X})$, an indicator function of the ice phase such that:

$$I(\mathbf{X}) = \begin{cases} 1 & \text{if } \mathbf{X} \text{ lies in the ice phase} \\ 0 & \text{if } \mathbf{X} \text{ lies in the air phase} \end{cases}$$

where $\mathbf{X} = (x, y, z)$ is a position vector within the sample. Homogeneous cubic subsets of size $n_{\max} = 700$ voxels were then extracted from the original images for further analysis.





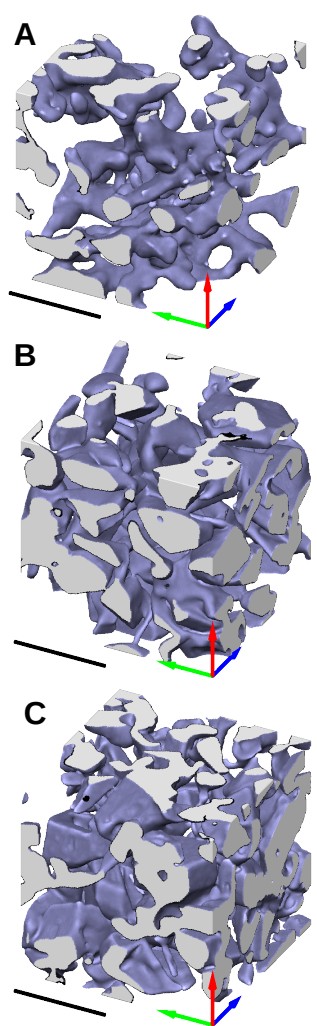

**Figure 2.** Microstructure of the samples S1, S2 and S3 as revealed by X-ray tomography. These visualisations correspond to subsets from the 3D images S1_B_1_7m (panel A), S2_B_1_7m (panel B) and S3_B_1_7m (panel C). The scale bar is 1 mm.

### 2.2.2 Chord length distribution

The *chord length distribution* (CLD), also called *chord length probability density*, is often used for the characterisation of binary porous media. It is based on a microstructure description in terms of random chords, i.e. iso-phase line segments, whose lengths $l$ are estimated by throwing virtual rays in random directions through the microstructure. The CLD $p^{(i)}(l)$ of phase $i$

5  denotes the probability $p^{(i)}(l)dl$ of finding a random chord of length between $l$ and $l + dl$ in phase $i$ (Torquato, 2002), thus giving us information on the thicknesses of the elements constituting the considered phase.





In the case of snow, which is known to be an anisotropic material with an orthotropic axis corresponding to the vertical (z) direction (see Calonne et al., 2011, 2012; Löwe et al., 2013; Calonne et al., 2014b, a; Wautier et al., 2015; Srivastava et al., 2016; Wautier et al., 2017), the CLD measured along a particular line depends on its direction. Assuming that the anisotropy is small, we consider the statistical characteristics of a sample in the three Cartesian directions. Namely, the chord lengths were

obtained by scanning the segmented images with "rays" along the $\{\mathbf{x}, \mathbf{y}, \mathbf{z}\}$-directions. As the resolution of the X-ray images $\Delta d$ is finite, the chord length $l$ takes 700 discrete values from $\Delta d$ to $d_{\mathrm{max}}=700\Delta d$ along every direction. The total number of chords of length $l$, $n_j^{(i)}(l)$, is then used to calculate the CLD of phase $(i)$ in each X-ray image $j$:

$$p_j^{(i)}(l) = \frac{n_j^{(i)}(l)}{\Delta d_j \sum_l n_j^{(i)}(l)} \tag{1}$$

All chords that cross the file borders are inherently dismissed by this estimation method; hence, no hypothesis on the exact

nature of the phase (air or ice) outside of the processed image is required. In what follows, the CLDs of the ice phase only are considered. The mean chord length $a_j$ and the characteristic function (CF), $L$ of the ice phase in X-ray image $j$ are obtained as:

$$a_j = \langle l \rangle_j \tag{2}$$

$$L_j(\alpha) = \langle \exp(-\alpha l) \rangle_j \tag{3}$$

where sign $\langle \rangle_j$ denotes averaging in image $j$. The overall characteristics of the sample are the average over its X-ray images:

$$X = \frac{1}{N} \sum_j X_j \tag{4}$$

where $X_j$ is any metric attributed to image $j$ and $N$ is a number of images of a sample. The sample anisotropy is estimated as :

$$\frac{2\langle l_z \rangle}{\langle l_x \rangle + \langle l_y \rangle} \tag{5}$$

where the subscript means the direction at which chords are taken.

### 2.2.3 Specific surface area (SSA)

The specific surface area was estimated using two different means: a *stereological* approach and a *voxel projection* method. Corresponding quantities calculated with these two methods are indicated with subscripts CLD and VP, respectively.





The stereological approach is directly based on the computation of the mean chord length $a$ of the ice phase obtained from the CLD analysis. Indeed, $a$ is uniquely related to the SSA and to the ice volumetric mass $\rho_{ice}$ by the following formula (see e.g. Torquato, 2002; Malinka, 2014):

$$\text{SSA}_{\text{CLD}} = \frac{4}{a\,\rho_{\text{ice}}} \tag{6}$$

The voxel projection estimation, denoted $\text{SSA}_{\text{VP}}$, corresponds to an approach developed by Flin et al. (2005, 2011). Based on a adaptive determination of the normal unit vector in each surface voxel, this method allows to obtain the surface area of the whole object of interest. The SSA can then be obtained after dividing the computed interface area $S_{\text{int}}$ by the associated volume. In the present implementation, a small improvement concerning the computation of the ice phase volume has been added. The VP method being based on the area estimation of the interface located within a particular voxel (Flin et al., 2005),

digitising divides an image into voxels of two types: ice and air, the interface voxels being systematically attributed to the ice phase. As a result, the total volume of ice voxels $V_{\text{VP}}$ slightly overestimates the true volume of the ice phase $V_{\text{ice}}$ by a quantity equal in average to a half volume of the interface voxels $V_{\text{int}}$. Therefore, $\text{SSA}_{\text{VP}}$ was computed with the formula:

$$\text{SSA}_{\text{VP}} = \frac{S_{\text{int}}}{\rho_{\text{ice}}\left(V_{\text{VP}} - \frac{V_{\text{int}}}{2}\right)} \tag{7}$$

This improved VP method has been validated against several data, including a series of calibrated spheres (see Appendix C).

**2.3   Optical measurements analysis**

As detailed in Dumont et al. (2010), to convert the spectrogonio-radiometer measurements into BRF values, we divide the measured radiance reflected from the snow surface by the radiance from a reference surface for which spectral albedo and BRF are known. For visible and near-IR wavelengths, the reference surface is a Spectralon® panel. For wavelengths longer than 2440 nm, an infragold® panel is used.

Let $R(\theta_i, \theta_v, \phi_i, \phi_v, \lambda)$ be the BRF of a sample under incident zenith angle $\theta_i$, viewing zenith angle $\theta_v$, incident azimuth $\phi_i$, viewing azimuth $\phi_v$ and wavelength $\lambda$. The reciprocity principle states that

$$R(\theta_i, \theta_v, \phi_v - \phi_i) = R(\theta_v, \theta_i, \phi_i - \phi_v) \tag{8}$$

In what follows, we use the anisotropy factor $\eta$ calculated using the following equation to quantify the anisotropy of the reflectance of a snow sample:

$$\eta(\lambda) = \frac{R(30°, 70°, 180°, \lambda) - R(30°, 70°, 0°, \lambda)}{R(30°, 70°, 180°, \lambda)} \tag{9}$$

Note that this parameter does not necessarily capture the position of the maximum and minimum reflectances especially in the visible wavelengths as discussed in Stanton et al. (2016).



### 2.4 Optical modelling

The model of snow reflectance used in this investigation is described in detail in Malinka (2014) and Malinka et al. (2016). The concept notion used in this model to characterise the snow microstructure is the mean chord length $a$, which can be seen as the effective snow grain size (Eq. 6) . The main quantity that characterises the optical properties of a snow layer is its optical thickness, $\tau$ , which can be calculated as:

$$\tau = \frac{\beta H}{a} \qquad (10)$$

where $\beta$ is the volume fraction of ice (1-$\beta$ is the snow porosity) and $H$ is the sample thickness.

The reflectance of snow in the visible range can also be strongly affected by light absorbing particles. Here we consider only one type of light absorbing impurities: soot (black carbon particles), which has a spectrally neutral absorption. As it is shown in Zege et al. (2008) the soot pollution is easy to take into account by replacing the imaginary part of the refractive index of ice, $\kappa_{ice}$, by the effective value $\kappa$ :

$$\kappa = \kappa_{ice} + 0.2 C_{st}, \qquad (11)$$

where $C_{st}$ is the relative soot concentration (soot-to-ice volume fraction).

The model described in Malinka et al. (2016) uses the three parameters (the optical thickness $\tau$, mean chord length $a$, and concentration of a contaminant with a predefined spectrum, $C_{st}$ in our case) to characterize in full the snow reflectance. The more general model of light scattering in porous materials (Malinka, 2014) characterizes the medium microphysical properties via the chord length distribution (CLD). The model of snow in Malinka et al. (2016) assumes exponential CLDs. This model was shown to be reliable in terms of reproducing the measured albedo spectra with the simulated ones in the visible and near IR range (from 350 to 1350 nm). However, in Malinka et al. (2016), there was no direct measurements of the snow microstructure characteristics to verify the model completely. In this work we verify the model by comparing the retrieved mean chord length with that measured directly by the X-ray tomography and investigate the bidirectional reflective properties in a rather wider spectral range (up to 3000 nm), addressing the effect of the μCT-measured CLD vs. the exponential one. To do so, we consider two configurations in the simulation:

- Assuming that the CLD is exponential with the mean values calculated directly from the 3D images (Tab. 2, col. 'Mean chord length, Aver.'). These simulations are labelled EXP hereinafter.

- Directly using the CLD calculated from the 3D images. The simulations are labelled μCT hereinafter.

For the simulations, we used the database of ice optical properties provided by Warren and Brandt (2008) except when another source is mentioned (Kou et al., 1993 and Grundy and Schmitt, 1998) in Sec. 3.4.





## 2.5 SSA retrieval from optical measurements

The optical model described in the previous section can be used to retrieve SSA from a measured albedo spectrum. First, the volume fraction of ice $\beta$ is calculated for a sample:

$$\beta = \frac{\rho}{\rho_{\mathrm{ice}}} \tag{12}$$

where $\rho$ is the average sample density taken from the 3D X-ray images. Then the optical thickness $\tau$ is calculated by Eq. 10 with an arbitrary starting value of the mean chord length $a$. After that, the new values of $a$ and $C_{st}$ are found that provide the best fit, in the least-squares meaning, of the measured albedo spectrum and the simulated spectrum for a given $\tau$ calculated using the model from Malinka (2014). The new $\tau$ is then calculated by Eq. 10 with the new value of a, and the procedure is repeated until the resulting values no longer change in the first five digits. It usually takes 2-3 iterations. The SSA and $a$ are
related with Eq. 6.

## 3 Results

Here we first provide a comparison of the characterization of the snow microstructure obtained with the different methodologies (manual measurements, optical measurements and X-ray tomography). In a second step, we evaluate the accuracy of the BRF measurements. The last two sections compare the BRF obtained from the model and from the measurements and evaluate the
impact of the microstructure on the BRF.

### 3.1 Snow microstructure

### 3.1.1 SSA and density

The SSA was obtained by various methods using both optical measurements and X-ray tomography. The optical methods include DUFISSS, ASSSAP, and retrieval from the albedo spectrum (Sec. 2.3). The retrieved values are shown in Table 2
together with the values calculated from the 3D X-ray images. The measured and retrieved albedo spectra of samples S1 and S2 are shown in Fig. 3.

    Figure 4 compares the SSA values obtained from the 3D images using either the chord length distribution calculation ($\mathrm{SSA_{CLD}}$) or the voxel projection method ($\mathrm{SSA_{VP}}$). It shows that the agreement between the two SSA values is good ($R^2$=0.994, RMSE=0.28 m$^2$ kg$^{-1}$). The lower SSA values are slightly overestimated via the CLD method, while the higher
ones are slightly underestimated. This agreement is to be compared with fig. 2 in Krol and Löwe (2016a).

    Figure 5 sums up all the density and SSA measurements performed on all the 3 samples.

    All three samples exhibit some variability both horizontally and vertically. The average density estimated from the 3D images is slightly higher than the manually measured one for S1 and S3 and slightly lower for S2. The average SSA calculated from the X-ray tomography is systematically lower than that estimated from DUFISSS and ASSSAP measurements and is in
perfect agreement with the SSA retrieved from the albedo spectra. These discrepancies might be inherent to the methodology,





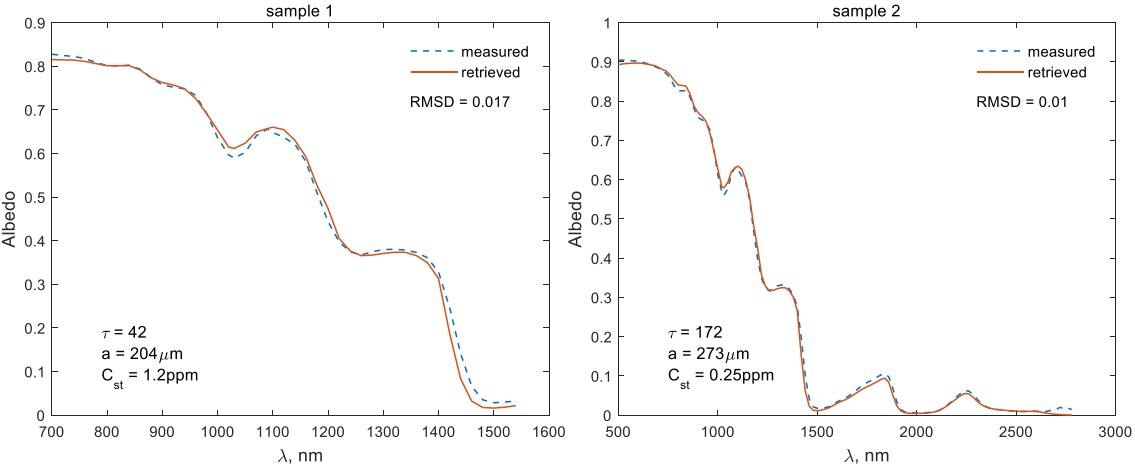

**Figure 3.** Measured and retrieved albedo spectra of samples S1 (left panel) and S2 (right panel).

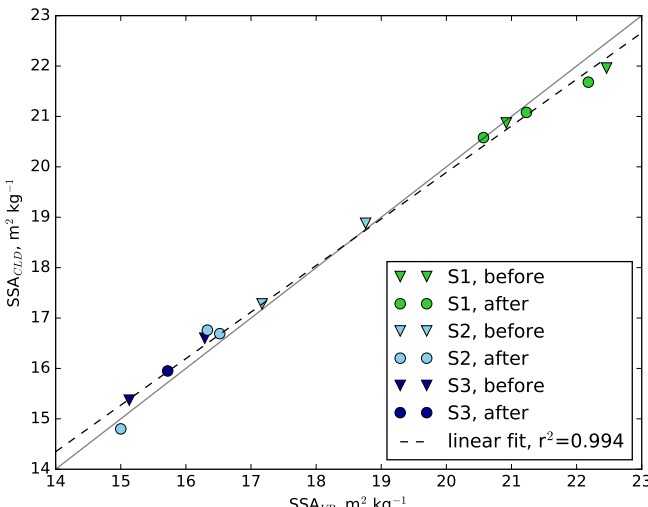

**Figure 4.** $SSA_{CLD}$ versus $SSA_{VP}$ in m$^2$ kg$^{-1}$ computed for all 3D images described in Table 1. The triangle and circles markers indicate that the samples were taken before and after the optical scans, respectively. Each sample is represented by a different colour. The linear fit is represented by the black dashed line.

but could also be linked to the variation of the properties inside the sample. Note that an X-ray tomography image has a very small size (of the order of several mm), the optical measurements with DUFISSS and ASSSAP uses surfaces with the size of about 5 cm, and the spectral albedo characterizes a sample as a whole.





**Table 2.** Sample microphysical properties calculated from the X-ray tomography images and retrieved from the spectral albedo. 'B' and 'A' refer to 'before' and 'after' the optical measurement, respectively.

| Sample | Density kg m$^3$ | | SSA, m$^2$ kg$^1$ | | a, $\mu$m | | a, μm | $C_{st}$, ppm | SSA, m$^2$ kg$^1$ |
| --- | --- | --- | --- | --- | --- | --- | --- | --- | --- |
| | | | VP | CLD | | | | | |
| | Value | Aver. $\pm\sigma$ | | | Value | Aver. $\pm\sigma$ | | | |
| S1_B_1_7m | 149 | | 20.92 | 20.87 | 209 | | | | |
| S1_B_1_10m | 136 | 141 ±12 | 22.46 | 21.96 | 199 | 205.5 ±5.5 | 204 | 1.21 | 21.4 |
| S1_A_1_10m | 122 | | 22.18 | 21.68 | 201 | | | | |
| S1_A_1_7m | 145 | | 20.57 | 20.58 | 212 | | | | |
| S1_A_2_7m | 153 | | 21.23 | 21.08 | 207 | | | | |
| S2_B_1_7m | 283 | | 17.17 | 17.28 | 252 | | | | |
| S2_B_2_7m | 248 | 262 ±13 | 18.76 | 18.88 | 231 | 260. ±23. | 273 | 0.246 | 16.0 |
| S2_A_1_7m | 256 | | 16.52 | 16.69 | 261 | | | | |
| S2_A_1_5m | 257 | | 16.33 | 16.76 | 260 | | | | |
| S2_A_1_10m | 265 | | 15.00 | 14.80 | 295 | | | | |
| S3_B_1_7m | 344 | | 15.13 | 15.37 | 284 | | | | |
| S3_B_2_7m | 354 | 340 ±16 | 16.29 | 16.60 | 263 | 273. ±11. | 288 | 0.232 | 15.1 |
| S3_A_1_7m | 322 | | 15.72 | 15.95 | 273 | | | | |

Figure 5 shows that S2 and S3 are denser and coarser — i.e. consist of larger grains— than S1 (almost new snow). The density after the BRF measurements is systematically higher than before and the SSA is systematically smaller (except for one of S3 measurements). This indicates that the snow microstructure as a whole has slightly evolved during the BRF measurements: snow has become denser and coarser (possibly because of sublimation in the cold room). This is to keep in mind when analysing the optical measurements. S1 exhibits larger changes in SSA than S2, and S2, in turn, demonstrates larger changes than S3. This is compatible with the fact that snow with lower density and higher SSA is subject to more rapid evolution (e.g. Flin et al., 2004; Carmagnola et al., 2014; Schleef et al., 2014). Also, the retrieval has shown that the snow taken in the mountains (S2 and S3) was much more pure: the soot concentration in it was 5 times less than that in the snow taken in the urban area (S1) (Tab. 2).



### 3.1.2 CLD analysis

Figure 6 compares the CLD of the three samples estimated from all the images. S1's CLD is narrower than S2's and S3s' CLD; however, all of them exhibit similar features. They have almost exponential tales for large $l$ and approach zero at $l = 0$. The latter fact means the ice-air interface does not have edges at the image resolution for the three snow samples, in contrast,

e.g., to a set of random polyhedra, which has a Markov property, i.e. the exponential CLD. Distributions of this type can be represented by the gamma distribution:

$$p(l) = \frac{l^{m-1}}{\Gamma(m)} \left(\frac{m}{a}\right)^m \exp\left(-\frac{m}{a}l\right) \tag{13}$$

where $a$ is an average, $\Gamma(m)$ is the gamma function, and $m$ is a shape parameter. The exponential distribution is a particular case of the gamma distribution with $m = 1$. Thus, the shape parameter $m$ indicates the deviation of a distribution of this type

from the exponential one. Fitted gamma distributions are also indicated in Fig. 6 (see below for the calculation of the value of $m$).

The CLD determines optical properties of a mixture not explicitly but via its Laplace transform (Malinka, 2014), which by definition is the characteristic function of the distribution:

$$L(\alpha) = \langle e^{-\alpha l} \rangle = \int_0^\infty e^{-\alpha l} p(l) dl \tag{14}$$

If the argument $\alpha$ is equal to the substance absorption coefficient, then the characteristic function $L(\alpha)$ describes the process of photon absorption while travelling along random chords within an absorbing material, in this case, ice (see Malinka, 2014). The Laplace transform of the gamma distribution (Eq. 13) is :

$$L(\alpha) = \left(1 + \frac{\alpha a}{m}\right)^{-m} \tag{15}$$

Figure 7 shows the characteristic functions calculated directly from the 3D images of S1 and S3 using Eq. 3 and their

approximations with the Laplace transforms (Eq. 15) of the gamma distribution with $m$ calculated by least-squares fitting. The Laplace transform of the exponential distribution ($m = 1$) is shown for comparison. The characteristic function of S2 is not shown because of the overlap with that of S3.

The characteristic functions calculated directly from the X-ray images at large values of argument $\alpha a$ could be distorted by the image discretisation. Their values are reliable if the error of the argument $\alpha a$ due to the discretisation is small. If the image

resolution is $\Delta d$ , then the condition $\Delta d \ll 1$ should hold true. For the resolution is 7 μm and the mean chord is about 200 μm, we have

$$\alpha \Delta d = \alpha a \frac{\Delta d}{a} \approx \frac{\alpha a}{30} \tag{16}$$





i.e. $\alpha \Delta d < 0.1$ for $\alpha a < 3$, so the μCT-measured characteristic functions presented in Figure 7 are not affected with the discretisation error.

The characteristic function in Eq. 3 has the power series expansion

$$L(\alpha) = \langle 1 - \alpha l + \frac{1}{2}(\alpha l)^2 + \mathcal{O}(\alpha^3 l^3) \rangle = 1 - \alpha a + \frac{\mu_2 \alpha^2}{2} + \mathcal{O}(\alpha^3 a^3) \tag{17}$$

where $\mu_2$ is the second moment of the CLD.

The exponential CLD has the Laplace transform from Eq. 15 with $m=1$, hence the expansion

$$L(\alpha) = 1 - \alpha a + \alpha^2 a^2 + \mathcal{O}(\alpha^3) \tag{18}$$

Comparing the second order term in Eq. 18 with that in the general expansion in Eq. 17, we find that the deviation of the CLD from the experimental law, in general, can be characterised by the value $\frac{\mu_2}{2a^2}$ (Krol and Löwe, 2016a) or its deviation from

unity, $\delta$ :

$$\delta = 1 - \frac{\mu_2}{2a^2} \tag{19}$$

For the gamma-distribution this value is expressed via the shape parameter as follows:

$$\delta = \frac{m-1}{2m} \tag{20}$$

The deviations from the exponential law $\delta$, calculated from m-values with Eq. 20 and directly from the CLD with Eq. 19 are

shown in Table 3. The values obtained in this experiment are slightly higher but consistent with the results of the analysis of Krol and Löwe (2016a).

**Table 3.** Values of $m$ and $\delta$ averaged across all the images.

| Snow type | m | $\delta$ | | |
|---|---|---|---|---|
| | | from m | from $\mu_2$ | from Krol and Löwe (2016a) |
| DF/RG (S1) | 2.40 | 0.292 | 0.269 ± 0.006 | 0.23-0.25 |
| FC/DH (S2) | 1.95 | 0.244 | 0.246 ± 0.007 | 0.12-0.21 |
| FC/DH (S3) | 1.94 | 0.242 | 0.232 ± 0.032 | 0.12-0.21 |

The deviation of the CLD from the exponential law matters, i.e. affects the optical properties, if the difference between Eq. 17 and Eq. 18 is not negligible, i.e. $\alpha^2 a^2 - \frac{\alpha^2 \mu_2}{2} = \alpha^2 a^2 \delta \sim 1$.

Thus, the absorption at which the deviation matters can be estimated as

$$\alpha \sim \frac{1}{a\sqrt{\delta}} \tag{21}$$

Provided $a \sim 200$ μm and $\delta \sim 0.25$, we get $\alpha \sim 10^4$ m$^{-1}$. So, we can expect that the true shape of the CLD will impact the optical properties only at very high absorptions. The absorption coefficient of ice in the range 500–2500 nm approaches such





values at 1500, 2000, and 2500 nm (see below). At these absorptions, for which $\alpha a \sim 2$, we can expect that the light penetration depth will not exceed the mean chord length $a$, treated as the effective snow grain size, so all the incident light will be absorbed within the skin layer — a layer at the surface with a thickness of "one grain". The reflectance will be completely determined with Fresnel reflection by crystal facets and refraction by fine grains in this skin layer. Under these conditions, the fine grains,

and therefore the CLD behaviour at small lengths, will play a more important role. The facet orientation at the surface can differ from that in deeper layers as well.

### 3.2 Optical measurements accuracy: Reciprocity principle

Figure 8 compares the reflectances obtained for optically equivalent geometries according to Eq. 8. For S1, Eq. 8 is really well verified. This is not the case for S2 and S3, especially in the visible range. The SSA temporal evolution during the BRF

measurements (see Fig. 5 in Sec. 3.1.1) cannot explain these discrepancies, since it would result in a decrease of reflectance in the measurements taken later, so that the red curves would be lower than the blue ones, but the contrary is observed. A possible explanation lies in the fact that the illumination pattern is larger in the sample holder for large incidence angles (e.g. 60°). Since the snow does not perfectly fill the sample holder, direct reflection of light incoming on the side of the sample holder to snow cannot be avoided, acting as an additional source of light and contributing to the reflected signal. This hypothesis is

reinforced when comparing other equivalent geometries ($\theta_i$=0°, $\theta_v$=30°) showing almost perfect agreement between the two reflectances. This indicates that reflectances measured for $\theta_i$=60° should be analysed accounting for the fact that they might be overestimated because of re-illumination, especially for low absorptive wavelengths.

This shows that the relative accuracy the measurements estimated at 1% in Bonnefoy (2001) is not reached for high illumination angles. In the following, the interpretation of the BRDF measurements is mostly restricted to wavelengths larger than

800 nm and —except in Sec. 3.4— to illumination angle of 30° to minimise this effect. For wavelengths larger than 800 nm, the effect of the limited thickness of the sample and of the presence of light-absorbing impurities (Warren, 1982) is also limited.

### 3.3 Model vs measurements

#### 3.3.1 Spectral variations

Figure 9 compares the measured and simulated spectral reflectances for all three samples. The results show that the simulated

reflectances agree generally well (absolute difference less than 0.02) except for several wavelengths ranges that exhibit systematic bias for all three samples. There is a little overestimation in the visible and near IR range up to 1400 nm, which is likely due to the geometry (see above). There are some peaks of the difference — 1000, 1200, and 1450 nm — where the reflectance, following ice absorption, has steep changes. However, the most pronounced differences are in the ranges 1400–1900 nm and 2150–2300 nm. These differences are especially pronounced for S1. We relate these differences to the difficulties of the mea-

surements of ice absorption in these ranges with high accuracy.

To examine this effect we simulate the snow reflectance in the most pronounced differences range 1400–1900 nm using the values of the complex refractive index of ice taken from different databases. Figure 10 compares the measured and the simulated





reflectances in this range for S1 and S2, using several databases of the complex refractive index of ice: Warren (1984), Kou et al. (1993), Grundy and Schmitt (1998), and Warren and Brandt (2008). For S1 both the incidence and observation angles are 60°. For S2 the observation angle is 60°, but the incident angle is taken 30°, because of the problems discussed in Section 3.2. As seen from Figure 9, the data from Grundy and Schmitt (1998) provides the simulated values closest to the measured ones and will be used in further simulations for this spectral region. However, as far as we understand, the additional careful measurements of the complex refractive index of ice in the IR range are still needed.





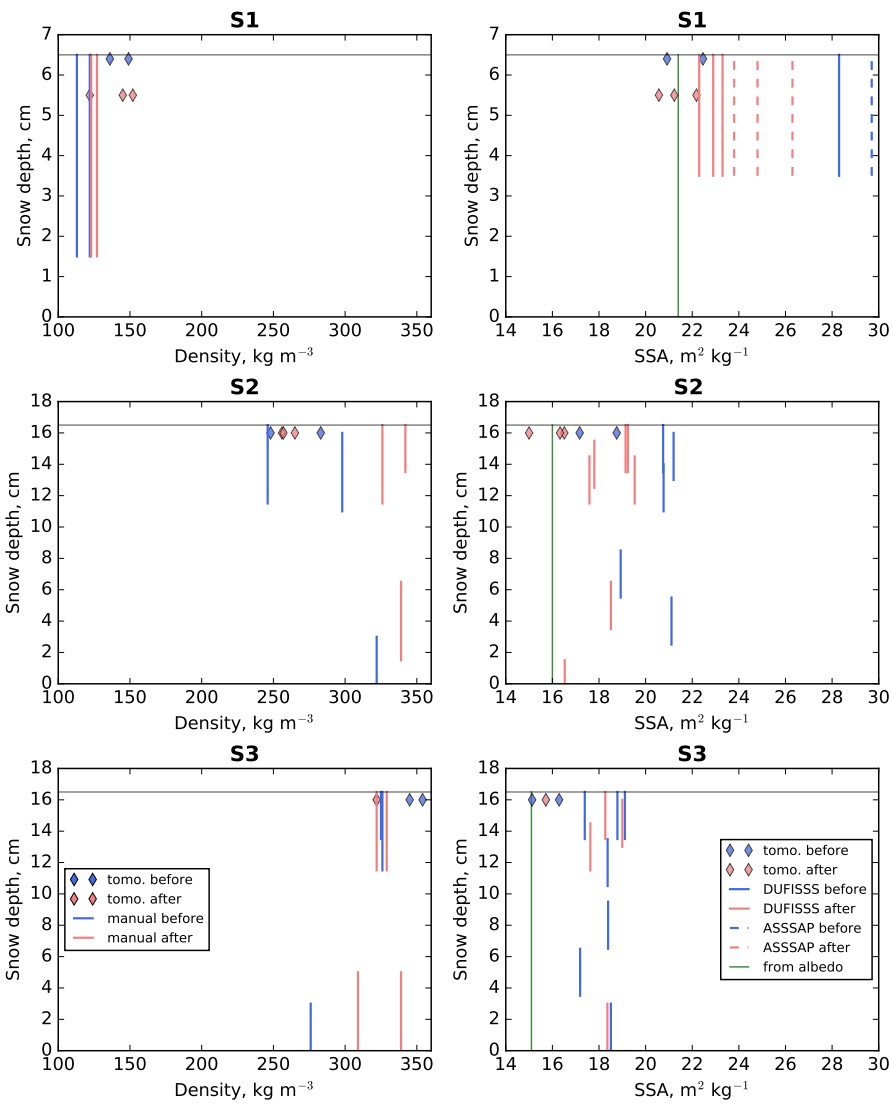

**Figure 5.** Estimated density (first column) and SSA (second column) at various sampling depth in S1, S2 and S3. Values estimated from 3D images are represented by diamonds. Values estimated from measurements (manual weighting for density, DUFISSS and ASSSAP for SSA) are represented by solid and dotted lines. The SSA retrieved from the spectral albedo is shown with vertical green lines. Values obtained before the optical scan are in blue and after in red. Note that the SSA obtained from 3D images represented in the figure are $SSA_{VP}$ values. The snow surface is represented by the horizontal grey line.





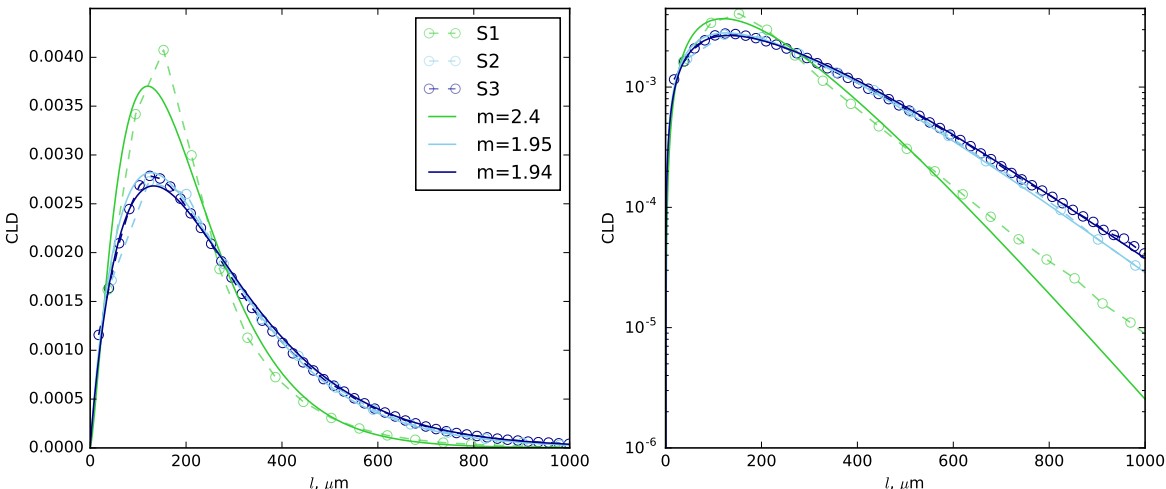

**Figure 6.** Left: CLDs in the three samples (circles and dotted curves) and their approximations with the gamma-distribution with the parameter $m$ (solid lines). Right : Same as the left panel but using the log scale. All the images from Tab. 1 have been used.

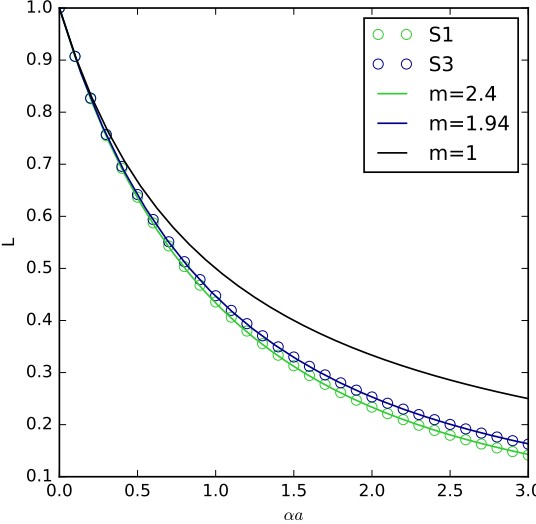

**Figure 7.** μCT-measured characteristic functions for S1 ans S3 and their approximations with the Laplace transform of the gamma distribution with different shape parameters $m$. All the images have been used for S1 and S3. S2 is not shown because of the overlap with S3.



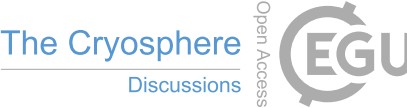

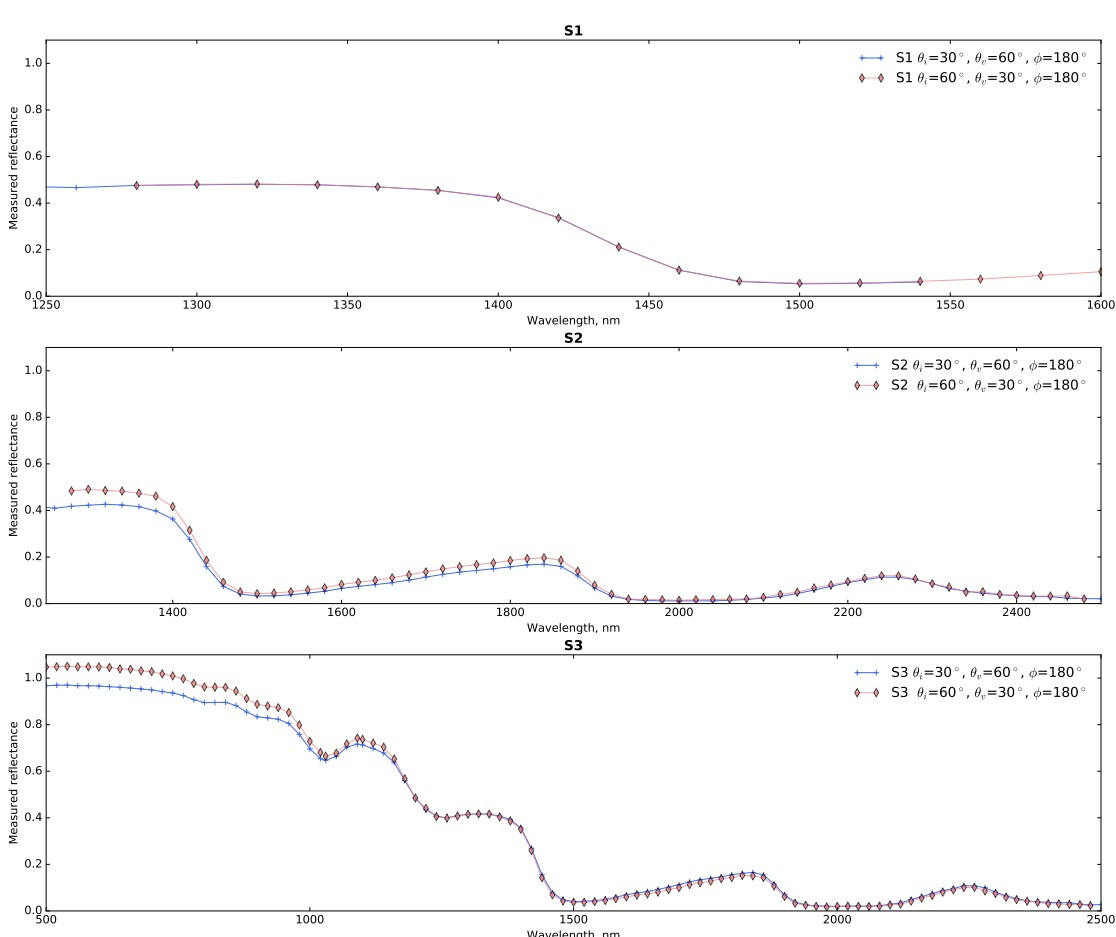

**Figure 8.** Measured reflectances of the 3 samples (each panel) obtained for optically equivalent geometries.



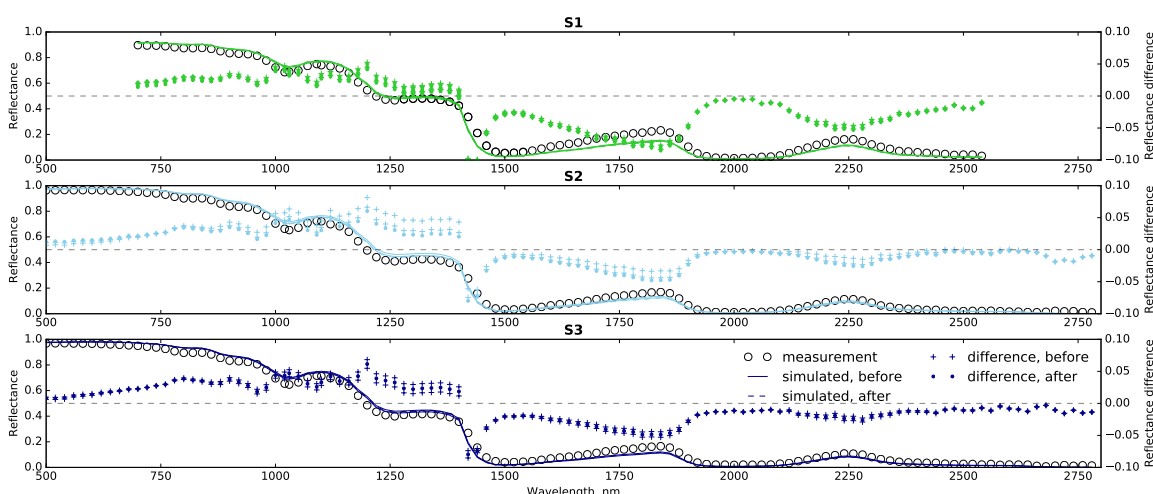

**Figure 9.** Measured (circles) and modelled (curves) BRF for the three samples (S1: upper panel, S2: centre panel, S3: lower panel) for $\theta_i$=30°, $\theta_v$=60° and $\phi_v$=180°. The solid and dashed curves correspond to the simulated reflectance using the snow microstructure measured before and after the optical measurements, respectively. All the simulations were performed with the exponential CLD. The differences between the measured and simulated reflectances are reported on the right y-axis with crosses and dots. The 3D images at 7 μm resolution, i.e. 3 images per sample, are used for the simulations.



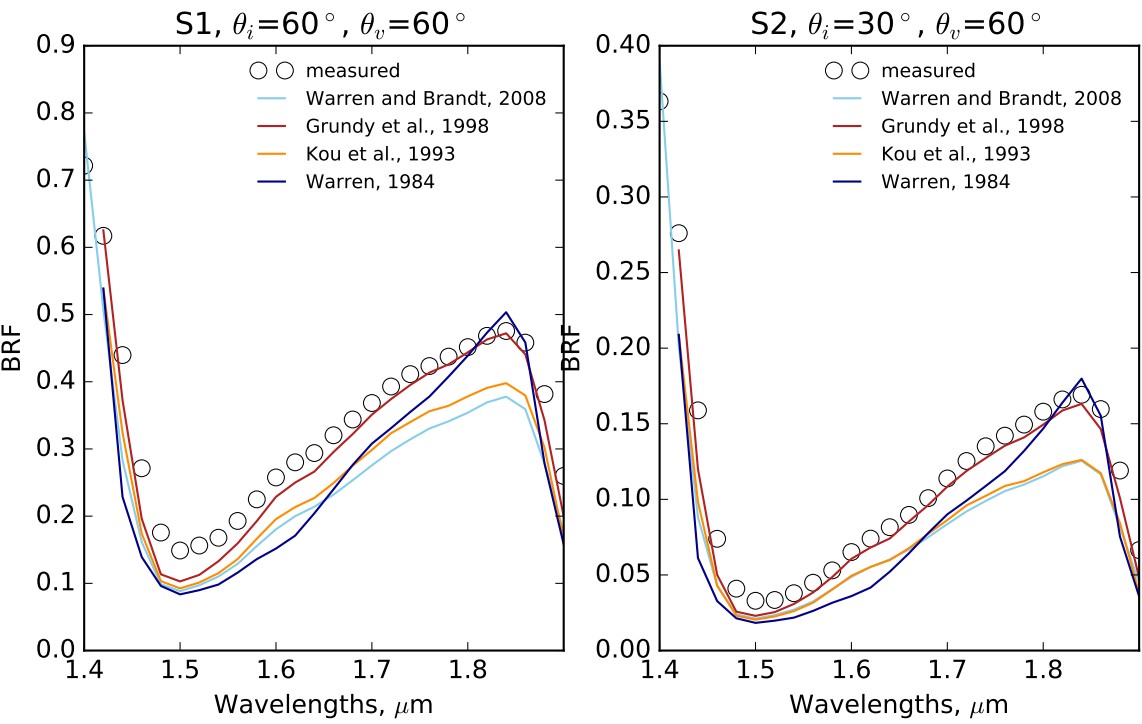

**Figure 10.** Measured (circles) and simulated (EXP, lines) reflectances in the principal plane for S1 (left) and S2 (right) in the 1400-1900 nm range. The incident zenith angles are $60°$ for S1 and $30°$ for S2. The viewing angle is $60°$ zenith and $180°$ azimuth. The BRF has been simulated using different ice refractive index databases: Warren and Brandt (2008) (light blue), Warren (1984) (dark blue), Grundy and Schmitt (1998) (red) and Kou et al. (1993) (orange).




### 3.3.2 Angular distribution

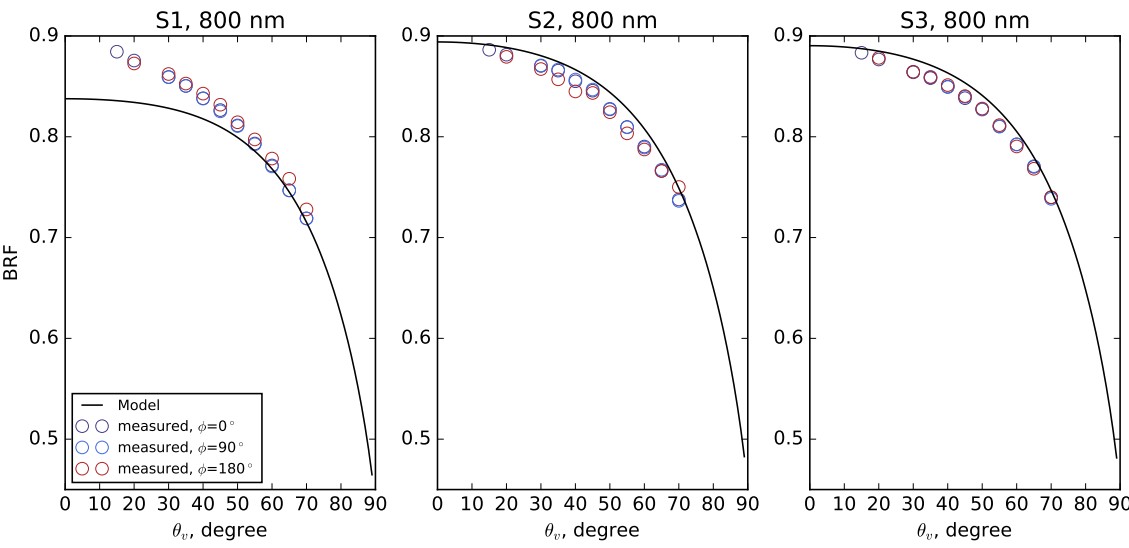

**Figure 11.** Measured (circles) and simulated (EXP) (curves) BRF of the three samples at vertical incidence at 800 nm.

Figures 11 and 12 compare the simulated and measured BRF at vertical incidence for 800 and 1300 nm. The simulated and measured data agree well, except for S1 at the observation directions close to vertical. Figure 13 shows the simulated and measured BRF at 30° incidence for a wide range of the observation angles. Except at 1500 nm, the simulated BRF generally

5     agrees with measured values. However, for high viewing zenith angle, typically higher than 45°, the simulated anisotropy is stronger than the measured one. These conclusions hold for S2 and S3 (see Figs. B3 and B4 in the appendices). At 1500 nm, the simulated BRF is lower than the measured values, as already reported in Fig. 9. The discrepancies between the simulated and measured BRF is generally greater than the spread in the simulation due to the variability of the snow microstructure (different images).

10     Figures 14, B1 and B2 (see Appendices) compare the measured and simulated angular distribution of reflectance for three wavelengths (1320, 800 and 1500 nm), for an incident zenith angle of 30° and for the two configurations of simulations (EXP and μCT).

Figure 14 is obtained for 1320 m (wavelength with medium absorption). It shows that (i) the angular pattern and reflectance magnitude are reasonably reproduced by the model and (ii) there are no detectable difference between the EXP and the μCT

15     simulations. It also shows that the higher reflectance values are obtained for S1 and the lower for S3, which is consistent with the SSA of the samples. The third line of Fig. 14 demonstrates that for the 3 samples, the simulated reflectance is overestimated in the forward scattering direction and underestimated in the backward direction, so at 1320 nm, the model seems to overestimate the anisotropy of the reflected radiation. The conclusions obtained at 1320 nm also stand at 800 nm (low absorption, Fig. B1). However at 1500 nm (high absorption, Fig. B2), the reflectance is underestimated almost twice for all angles and all samples

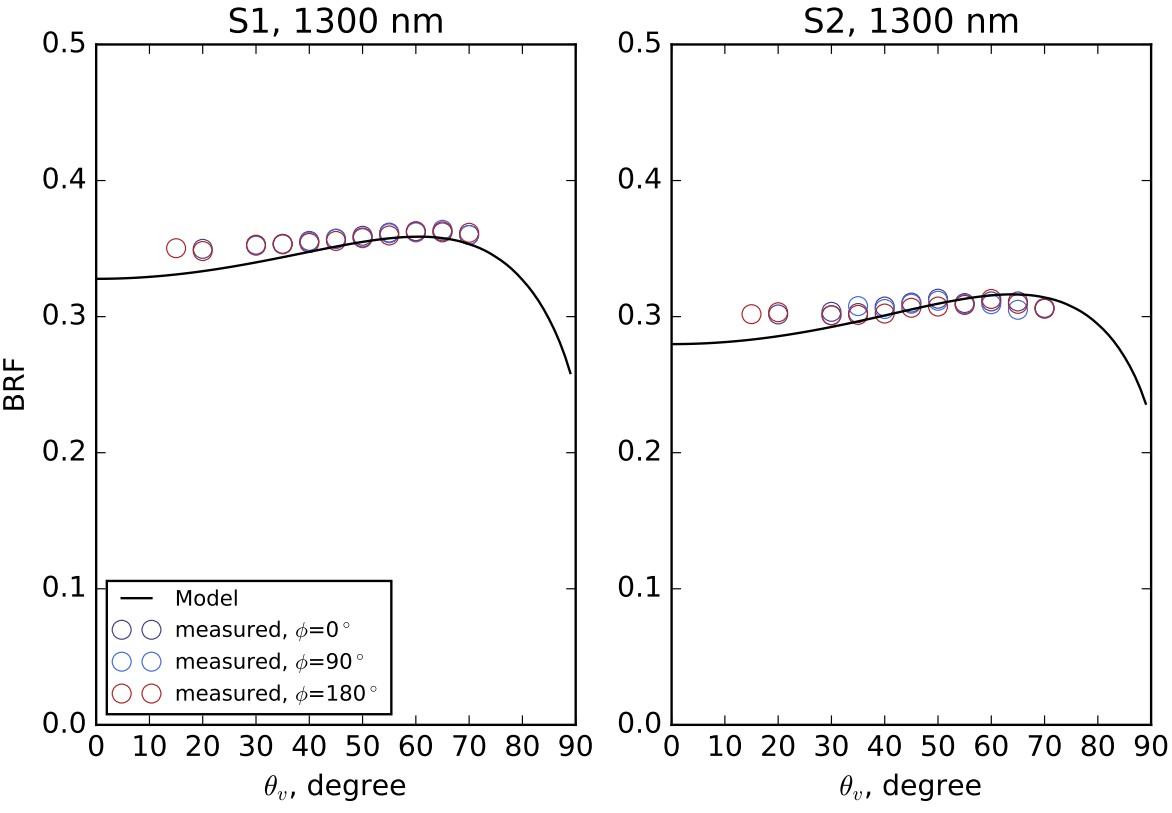

**Figure 12.** Measured (circles) and simulated (EXP, curves) BRF for S1 (left) and S2 (right) at vertical incidence at 1300 nm.

in the simulations. The reflectances for this wavelength are very low and can also be affected by experimental uncertainties. Stronger differences in reflectances at 1500 nm are noticeable between the samples and also between the EXP and μCT simulations.

Figure 15 compares the anisotropy of the reflectance measured and simulated quantified by the anisotropy parameter $\eta$ (Eq. 9). It shows that $\eta$ is higher in the simulations than in the measurements, except for highly absorptive wavelengths (around 1500 and 2000 nm), where the differences between the samples are higher in the measurements than in the model. Also, the EXP and μCT simulations only differ for these highly absorptive wavelengths.

### 3.4 Results at 60° illumination angle

Figure 16 compares the measured and simulated reflectance in the principal plane (forward direction) for an incident angle of 60° at several wavelengths. At high viewing angles, for all wavelengths and despite the lower SSA, S3 exhibits higher reflectance than S2 and S1. The higher the absorption, the more pronounced is this effect, so it might be related to the facet orientation at the surface. At 1280 nm, the simulations and the measurements agree well, with a slight overestimation of



**Figure 13.** BRF measured (circles) and simulated with the exponential (curves) and μCT-retrieved (dashes) CLDs for S1 at 700, 1030, 1300 and 1500 nm. The incident zenith angle is 30°. The different colours correspond to the azimuth. The μCT-retrieved CLDs are averaged across all of the S1 X-ray images.

the reflectance at high viewing angles. For the other wavelengths, the model generally underestimates reflectance. The use of alternative ice refractive index values (Grundy and Schmitt, 1998) improves the simulations at 1420 and 1500 nm. The difference between the μCT and EXP simulations are only noticeable for high viewing angles and high absorption. Under these conditions, accounting for the CLD shape retrieved from μCT leads to a decrease in the calculated reflectance in comparison

5  with the simulations with exponential CLD.



**Figure 14.** Measured (first line) and simulated (second line) reflectances for the three samples (three columns) at 1320 nm. The incident zenith angle is 30°. On the polar plot, the polar angle represents the azimuth and the radius is proportional to the viewing angles. The dotted circles represent viewing angles of 30, 60 and 70°. The two lower lines represent the difference between the simulated and measured reflectance for the exponential and μCT-retrieved CLDs. The images used for the simulations are S1_E_1_7m, S2_B_1_7m and S3_B_1_7m.
.





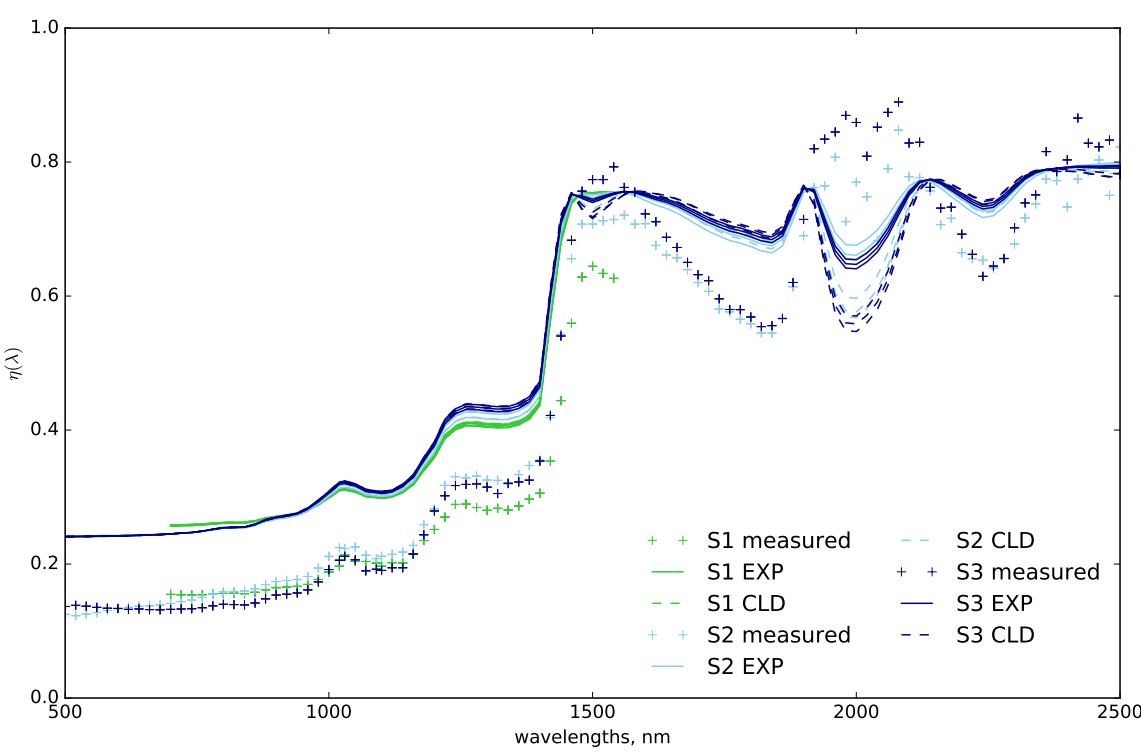

**Figure 15.** Measured (crosses) and simulated (dotted curves for µCT and solid curves for EXP) values of parameter $\eta$ (Eq. 9) obtained for the 3 samples. All the images were used for the simulations.



**Figure 16.** Measured (markers) and simulated (lines) reflectance in the principal plane (forward direction) for an incident zenith angle of 60° at 1280, 1420, 1500 and 2000 nm. Plain lines correspond to EXP and dotted lines to μCT simulations. All images have been used as inputs for the simulations. The red and orange lines correspond to simulation obtained for S1 with refractive index from Grundy and Schmitt (1998) and Kou et al. (1993). For all the other simulations we used refractive index from Warren and Brandt (2008) .





## 4   Discussions and conclusions

This study presents a dataset that combines the snow bidirectional reflectance over the 500-2500 nm range with different illumination geometries to 3D images of snow using X-ray tomography, which allows analysing the snow microstructure, e.g. its SSA and density. The SSA is calculated using two different methods: the voxel projection method and the method based on
the ice chord length distribution (CLD). The comparison between the two SSA values are in excellent agreement ($R^2$=0.994). The SSA and mean chord values computed from 3D images are generally lower than those obtained via DUFISSS and ASSSAP optical methods and in excellent agreement with those retrieved via spectral albedo. The comparison between density values obtained via the 3D images and via manual measurements exhibits slightly higher density for 3D images, which might relate to the heterogeneity of the samples and to the segmentation process applied to the images (see fig. 12 in Hagenmuller et al.,
2013) . The analysis of the 3D images has shown that the CLD approach zero at $l = 0$ both in the DF/RG snow and in the snow after long evolution in temperature gradient conditions. This means that the air-ice interface is smooth and has no sharp edges even though faceted crystals are present, at least for the images considered in this study. The reason for this is the continued curvature driven metamorphism of snow, which begins already during a snowfall (Flin et al., 2003; Krol and Löwe, 2016b).

The analysis of the characteristic functions of random chords in the snow phase calculated directly from the 3D images has
shown that the random chords obey the gamma distribution with the shape parameter $m$, equal to 2.40, 1.95, and 1.94 for S1, S2, and S3, respectively. The deviation of the distribution from the exponential one is the lowest for the more faceted crystals (sample 3) and the higher for DF/RG (sample 1). Its values, however, largely vary for the same macroscopic sample probably due to both the temporal evolution and the snow heterogeneity. The deviation values are slightly higher but in the same range as obtained by Krol and Löwe (2016a).

The comparison between the simulated and measured reflectance under a specific geometry ($\theta_i$=30° or 60°, $\theta_v$=60° and $\phi_v$=180°) shown in Fig. 9 demonstrated that the simulated values generally agree well (absolute difference lower than 0.03) over the whole spectrum 500-2500 nm and that for this geometry the impact of the CLD is small. Systematic differences are found in several wavelength ranges. Such discrepancies have already been reported in several studies (e.g. Carmagnola et al., 2013) and attributed to uncertainties in the values of the ice complex refractive index from Warren and Brandt (2008). The
use of alternative databases, especially the one from Grundy and Schmitt (1998) improves noticeably the agreement in the range 1400–1500 nm as shown by Fig. 10. A more extensive comparison of the simulated and measured reflectance angular distribution at 30° illumination angle shows that the BRF predicted by the model is in good agreement with the measured one. Spectral changes in the angular distribution are well reproduced. The impact of the CLD shape on the simulation is only detectable for absorptive wavelengths (1500 nm) and high viewing zenith angles. The anisotropy of the reflectance, quantified
by the relative difference between the reflectances measured at 70° zenith angle in the forward and backward direction in the principal plane, is systematically overestimated in the simulations.This effect was reported by many authors and commonly attributed to the surface roughness (Warren et al., 1998; Painter and Dozier, 2004; Hudson et al., 2006; Jin et al., 2008; Carlsen et al., 2020), which was not accounted for in our simulations and generally leads to less anisotropic angular patterns. However, the opposite effect is observed in the absorption bands around 1500 and 2000 nm, where the larger discrepancies are found





between the samples. For these wavelengths the model underestimates the intensity of forward scattering in the principal plane. Despite the lower SSA, sample 3 exhibits the higher reflectance at 70 and 75° zenith viewing angles. This might be related to the preferred upward orientation of the facets at the surface of this sample.

To sum up, the results exhibit two different trends for small and medium/long optical paths in the snow.

– For long/medium optical paths, the model predicts no impact of the CLD shape on the reflectance and is in good agreement with the measurement. The anisotropy is slightly overestimated in the simulations. A possible explanation can be related to the surface roughness which is not accounted for in the simulations and generally leads to less anisotropic patterns (Hudson et al., 2006). Differences between the reflectance of the three samples in the IR are well reproduced using the exponential CLD.

– The picture is different for the small optical paths, which refer to the high absorptive wavelengths and to oblique illumination and/or viewing. The model predicts an impact of the CLD. The use of the µCT-retrieved CLD instead of the exponential distribution leads to a decrease in the intensity of the forward scattering peak in the principal plane. The higher forward scattering intensities are observed for the more faceted crystals (S3) with facets upward. S3 also depicts the CLD closest to the exponential one. So, this behaviour is consistent with the model prediction. Several hypotheses

can be drawn to explain the model/measurements discrepancies. The first hypothesis is related to the imaginary part of the ice refractive index, to which the reflectance, including albedo, is very sensitive, but it is probably not sufficient to explain the different behaviours of the bidirectional reflectance distribution shown in Figs. 15 and 16. The second possible reason is the anisotropy of snow, which was noted by several authors (Calonne et al., 2012; Löwe et al., 2013) and for our samples ranged from 0.9 to 1.07, being calculated with Eq. (5). The last but most probable reason is the 'skin

effect.' For small optical paths, the light penetration depth is really small and probably less than a mm (e.g. Mary et al., 2013), so the role of the skin layer, where the grains could be finer and have a preferable facet orientation, is crucial. This is also corroborated by the fact that S2 and S3 have only a small difference in SSA, measured in deeper layers, but a strongly different behaviour in reflectance.

The effect of crystal shape and SSA on snow BRF is intricate (e.g. Dumont et al., 2010; Stanton et al., 2016). The dataset

presented in this study is limited to only three snow samples, so no statistically robust conclusions on the effect of crystal shape on reflectance can be drawn. The measurements, however, shows that the faceted crystals exhibit more anisotropic reflectance than fragmented particles/rounded grains as observed in Stanton et al. (2016) and that the anisotropic behaviour is further reinforced if facets are orientated upward. More quantitative relationships between crystal shape and the BRF would require a much larger number of snow samples analysed. One way of addressing this problem is to take advantage of ray tracing models

such as those developed by Kaempfer et al. (2007); Picard et al. (2009); Petrasch et al. (2007), which can be run directly on 3D images of the microstructure.

To conclude, this unique dataset combining X-ray tomography imaging of snow microstructure and high accuracy measurements of snow BRF was used to demonstrate that:

    – Faceted crystals exhibit a more anisotropic reflectance than fragmented particles.





– Malinka et al. (2016) model can be used to accurately simulate the snow BRF using SSA as input. The model, however, slightly overestimates the snow reflectance anisotropy. Even so, the mean chord length and SSA retrieved from the albedo spectrum and those measured by the X-ray tomography are in excellent agreement. As far as we know, such a successful comparison of the mean chord and SSA of snow retrieved from optical and CT measurements has been obtained for the first time.

– Other characteristics of snow microstructure besides SSA, e.g. the CLD shape, impacts the angular reflectance of snow for high ice absorption and oblique viewing and illumination.

## 5   Data availability

The BRF dataset will be made available in the PANGAEA database along with the CLDs and the simulation results. All simulations are based on the equations from Malinka (2014) and Malinka et al. (2016).

*Author contributions.*  M. Dumont led and wrote the study. F. Flin conducted the X-ray tomography measurements and the 3D images analysis. A. Malinka performed the analysis of the CLDs and their links to the snow reflectance and made the appropriate simulations. O. Brissaud conducted the BRF measurements together with M. Dumont. P. Hagenmuller and P. Lapalus performed the 3D image processing. B. Lesaffre, A. Dufour and N. Calonne prepared the snow samples with the two first authors and contributed to the X-ray tomography measurements. S. Rolland du Roscoat and E. Ando contributed to the X-ray tomography measurements.

*Acknowledgements.*  CNRM/CEN is part of Labex OSUG@2020 (ANR-10-LABX-0056). The 3SR lab is part of the LabEx Tec 21 (Investissements d'Avenir, grant agreement ANR-11-LABX-0030). This work was partly funded by ANR grants DigitalSnow (ANR-11-BS02-009) , EBONI (ANR-16-CE01-0006) and MiMESis-3D (ANR-19-CE01-0009), the State Research Program 'Photonics, Opto- and Microelectronics', and the National Academy of Sciences of Belarus. M.D. has received funding from the European Research Council (ERC) under the European Union's Horizon 2020 research and innovation programme (grant agreement No 949516, IVORI). The authors are thankful to Q. Libois, G. Picard, H. Löwe and Q. Krol for fruitful discussions on the paper. P. Charrier, J. Roulle, P. Puglièse and L. Pézard are also thanked for their help in the experimental part of the study.





## Appendix A: Overview of the optical measurements

**Table A1.** Geometry and spectral range of S1 optical measurements

| Name | $\theta_i$ | $\phi_{view}$ | $\theta_{view}$ | spectral range (nm) | begin | end |
|---|---|---|---|---|---|---|
| 2012_run1 | 30 | 180 | 70,60,50,45,40,30,20,10,0 | 1500,1520 nm | 03-27 15:23 | 03-27 15:31 |
| 2012_run2a | 30 | 0,30,60,90,120,150,180 | 0,15,30,45,60,70 | 700-1540 (20 nm) | 03-27 15:59 | 03-28 00:20 |
| 2012_run3a | 60 | 180 | 70,65,60,50,40,30,20,10,0 | 1280-2500 (20 nm) | 03-28 08:48 | 03-28 11:55 |
| 2012_run4a | 0 | 0,90,180 | 15,20,30,35,40,45,50,55,60,65,70 | 800, 810 | 03-28 14:14 | 03-28 14:39 |
| 2012_run5a | 0 | 0,90,180 | 15,20,30,35,40,45,50,55,60,65,70 | 1300, 1310 | 03-28 15:13 | 03-28 15:38 |
| 2012_run6a | 0 | 0,90,180 | 15,20,30,35,40,45,50,55,60,65,70 | 900-1090 (10 nm) | 03-28 16:10 | 03-28 19:43 |
| 2012_run7a | 30 | 180 | 70,60,50,45,40,30,20,10,0 | 1500, 1520 | 03-29 08:17 | 03-29 08:24 |

**Table A2.** Geometry and spectral range of S2 optical measurements

| Name | $\theta_i$ | $\phi_{view}$ | $\theta_{view}$ | spectral range (nm) | begin | end |
|---|---|---|---|---|---|---|
| 2013_run1 | 30 | 0,30,60,90,120,150,180 | 0,15,30,45,60,70 | 1500 | 03-20 17:11 | 03-20 17:28 |
| 2013_run2a | 30 | 0,30,60,90,120,150,180 | 0,15,30,45,60,70 | 500-2780 (20 nm) | 03-20 17:38 | 03-21 15:41 |
| 2013_run1_apM | 30 | 0,30,60,90,120,150,180 | 0,15,30,45,60,70 | 1500 | 03-21 17:51 | 03-21 18:08 |
| 2013_run3a | 60 | 180 | 0,10,20,30,40,50,60,65,70 | 1280-2480 (20 nm) | 03-22 08:19 | 03-22 11:17 |
| 2013_run4a | 0 | 0,90,180 | 15,20,30,35,40,45,50,55,60,65,70 | 800 | 03-22 11:47 | 03-22 12:02 |
| 2013_run5a | 0 | 0,90,180 | 15,20,30,35,40,45,50,55,60,65,70 | 1300 | 03-22 14:02 | 03-22 14:17 |





**Table A3.** Geometry and spectral range of S3 optical measurements

| Name | $\theta_i$ | $\phi_{view}$ | $\theta_{view}$ | spectral range (nm) | begin | end |
|---|---|---|---|---|---|---|
| 2013_run1_ech2 | 30 | 0,30,60,90,120,150,180 | 0,10,20,30,35,40,45,50,55,60,65,70 | 1500 | 3-22 15:17 | 3-22 15:50 |
| 2013_run2a_ech2a | 30 | 0,30,60,90,120,150,180 | 0,10,20,30,35,40,45,50,55,60,65,70 | 500-2780 (20 nm) | 3-22 16:15 | 3-24 15:16 |
| 2013_run3a_ech2a | 30 | 0,30,60,90,120,150,180 | 0,10,20,30,35,40,45,50,55,60,65,70 | 1500 | 3-25 09:15 | 3-25 09:47 |
| 2013_run4a_ech2a | 60 | 180 | 0,10,20,30,40,45,50,60,65,70,75 | 500-2480 (20 nm) | 3-25 10:47 | 3-25 17:25 |
| 2013_run5a_ech2a | 0 | 0,90,180 | 15,20,30,35,40,45,50,55,60,65,70 | 800, 810 | 3-25 17:53 | 3-25 18:18 |
| 2013_run6a_ech2a | 0 | 0,90,180 | 15,20,30,35,40,45,50,55,60,65,70 | 900-1080 (10 nm) | 3-25 18:30 | 3-25 21:53 |
| 2013_run7b_ech2a | 30 | 180 | 75 | 1000-2780 (20 nm) | 3-26 10:23 | 3-26 10:24 |
| 2013_run8a_ech2a | 30 | 0,30,60,90,120,150,180 | 0,10,20,30,35,40,45,50,55,60,65,70 | 1500 | 3-26 11:29 | 3-26 12:01 |



## Appendix B: Model vs measurements



**Figure B1.** Measured (first line) and simulated (second line) reflectances for the three samples (three columns) at 800 nm. The incident zenith angle is 30°. On the polar plot, the polar angle represents the azimuth and the radius is proportional to the viewing angles. The dotted circles represent viewing angles of 30, 60 and 70°. The two lower lines represent the difference between the simulated and measured reflectance for the exponential and $\mu$CT-retrieved CLDs. The images used for the simulations are S1_E_1_7m, S2_B_1_7m and S3_B_1_7m.





**Figure B2.** Measured (first line) and simulated (second line) reflectances for the three samples (three columns) at 1500 nm. The incident zenith angle is 30°. On the polar plot, the polar angle represents the azimuth and the radius is proportional to the viewing angles. The dotted circles represent viewing angles of 30, 60 and 70°. The two lower lines represent the difference between the simulated and measured reflectance for the exponential and $\mu$CT-retrieved CLDs. The images used for the simulations are S1_E_1_7m, S2_B_1_7m and S3_B_1_7m.





**Figure B3.** BRF measured (circles) and simulated with the exponential (curves) and $\mu$CT-retrieved (dashes) CLDs for S1 at 700, 1030, 1300 and 1500 nm. The incident zenith angle is $30°$. The different colours correspond to the azimuth. The $\mu$CT-retrieved CLDs are averaged across all of the S2 X-ray images.





**Figure B4.** BRF measured (circles) and simulated with the exponential (curves) and $\mu$CT-retrieved (dashes) CLDs for S1 at 700, 1030, 1300 and 1500 nm. The incident zenith angle is $30°$. The different colours correspond to the azimuth. The $\mu$CT-retrieved CLDs are averaged across all of the S3 X-ray images.




# Appendix C: Evaluation of SSA on calibrated spheres using different methods

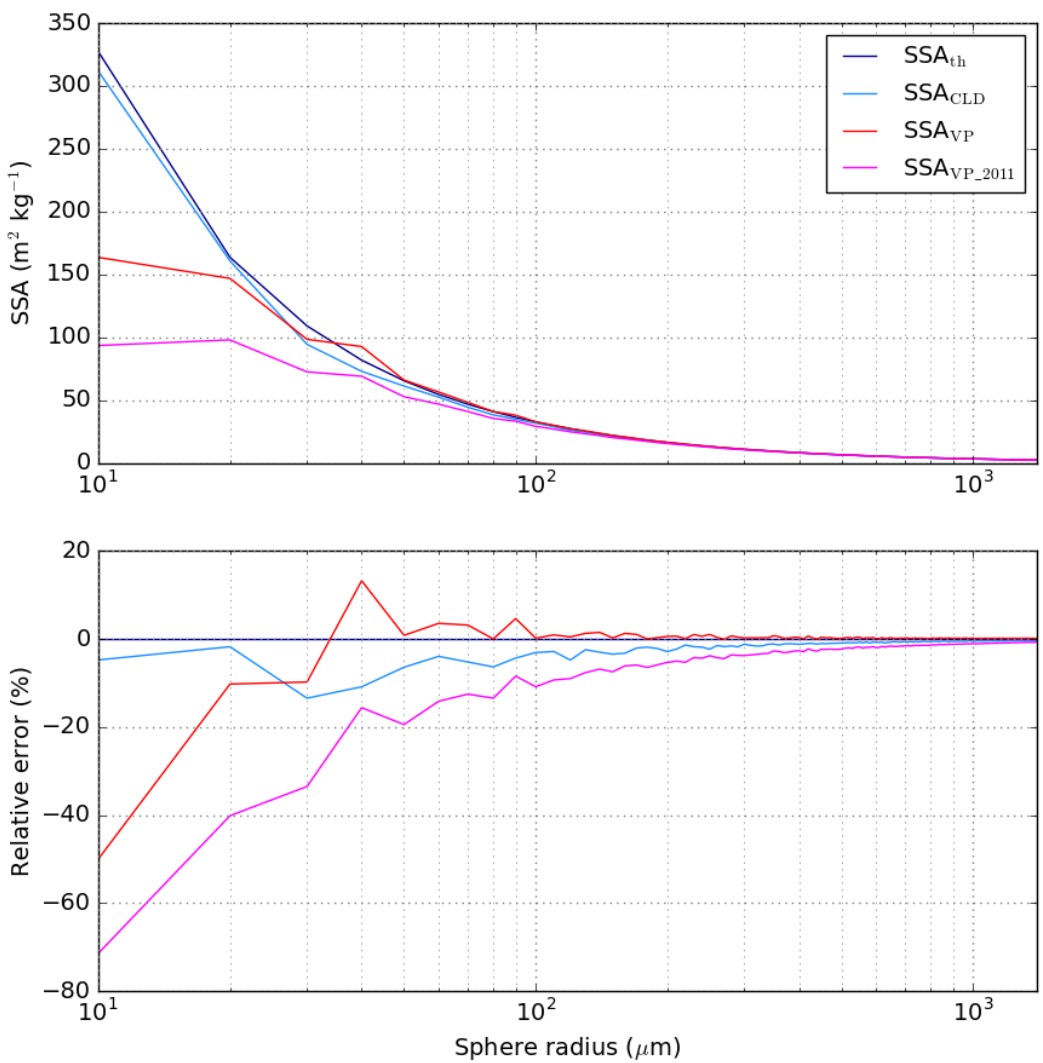

**Figure C1.** SSA estimated from different methods for spheres and their relative error compared to $SSA_{th}$. Sphere radius ranges from $R = 1$ to $R = 140$ voxel (1 voxel was arbitrarily chosen to correspond to 10 μm) - Overview in log scale. Note that here, only individual spheres (perfectly isotropic objects) are considered, so we have $SSAM = SSA_{th}$.



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
