# Peer review of "Experimental and model-based investigation of the links between snow bidirectional reflectance and snow microstructure"

_The Cryosphere, 2021_

## Author Response (AR1)

**Author response to comments from referee #1 on tc-2021-53**

Referee comments are in black. Author responses are in blue and proposed changes in the manuscript are highlighted in **bold blue**.

**Comments from referee#1**

This study presents a new observational dataset and the corresponding model simulation to examine snow microphysics and its impacts on snow radiative properties. Such work is desirable and especially crucial for remote sensing retrievals and data assimilation over the snow-covered regions and is also critical for global climate modelings constrained by reanalysis data. The experiment is properly designed and the discussion is well presented. The reviewer only has some minor questions regarding the sample treatment.

The authors thank the referee for the encouraging feedback. All comments have been accounted for and detailed responses are provided below.

Section 2.1:
"S3 is taken from the same temperature gradient experiment as S2 except that it was turned upside-down so that the grain orientation is changed by 180." Why did you flip sample S3?

Under temperature gradient, the facet formation is oriented toward the warmer side of the snow layer. e.g., when the temperature gradient is pointing downward as it is generally the case in nature, the facets tend to form on the downward surfaces while the upward surfaces stay more rounded (see for example figures 5 and 8 in Calonne et al., 2014a). As a consequence, by flipping S3, the faceted surfaces are oriented upward instead of downward in S2. This flip was thus done to investigate the effect of facet orientation on BRF, the other properties (SSA, density) being relatively close for S2 and S3.

To provide more details about this specific goal, we modified p4 line 1 as follows: "… is changed by 180°. **Under temperature gradient, the facet formation is oriented toward the warmer side of the snow layer. For instance, when the temperature gradient is pointing downward as usually the case in nature, the facets tend to form on the downward surfaces while the upward surfaces stay more rounded (see e.g. Figs. 5 and 8 in Calonne et al., 2014a). As a consequence, by flipping S3, the faceted surfaces are oriented upward instead of downward in S2. This was done to investigate the effect of facet orientation on BRF. "**

Section 2.1.1
"a 7 cm thick snow layer was collected on a 60x60 cm2 styrodur plate after a snowfall close to the lab and stored for 3 weeks in isothermal conditions at -20 C (Fig. 1A)." Why did the authors store the snow for 3 weeks before measurements? Would snow morphology alter during the storage time?

The snow was stored under isothermal conditions since our goal was to obtain a DF/RG sample, i.e., a relatively recent snow sample, but which was sufficiently metamorphosed to exhibit only smooth and rounded shapes. Due to the limited availability of the instrument used for the optical scan, we also had to control the snow evolution to reach the target morphology at the time of experiment. We first started at -20°C for which the changes are relatively slow (see e.g. Kaempfer and Schneebeli, 2007). Closer to the optical experiment, we checked the microstructure and imposed isothermal conditions at -10°C, to get the wanted DF/RG morphology.

Page 4 line 11 was thus modified as follows: "…after a snowfall close to the lab and stored for 3 weeks in isothermal conditions at -20 C. **It then stayed 3 days at -10°C to reach the DF/RG state (Fig. 1A). The objective of this imposed isothermal metamorphism was to obtain a relatively recent snow sample, but with smooth and rounded shape, and that can be resolved at the pixel size we could access with the tomograph (between ~ 6 and 12 µm)**. "

Kaempfer, T. U., and Schneebeli, M. (2007), Observation of isothermal metamorphism of new snow and interpretation as a sintering process, J. Geophys. Res., 112, D24101, doi:10.1029/2007JD009047.

**Additional modifications:**

Line 30 p 3: "S1 **corresponds to** decomposing and fragmented particles/rounded grains (DF/RG) according to the classification of Fierz et al. (2009)."
Line 1 p 13: "Figure 5 shows that S2 and S3 are denser and coarser—i.e. consist of larger grains—than S1 (**decomposing and fragmented particles/rounded grains**)."

"A vertical temperature gradient of 19.4 Cm−1 was applied inside the box with a mean temperature of -4 C". Could the authors provide more information on why pick this temperature gradient and -4 degree C? Was this tested in a previous experiment? If so, please provide some references here.

The objective of this experiment was to produce large faceted crystals, with relatively simple structures, exhibiting a clear asymmetry between their upper (rounded) and their downer (faceted) sides. Having performed experiments that previously led to this kind of results (Flin and Brzoska, 2008 and Calonne et al 2014a), we used them as guidelines to reach our purposes.

Page 4 line 15 was thus modified as follows: "…of -4°C. **Such conditions produce simple structures of large and regular faceted crystals in a reasonable amount of time (Flin and Brzoska, 2008 and Calonne et al 2014a).**"

REF: Flin, F., & Brzoska, J. (2008). The temperature-gradient metamorphism of snow: Vapour diffusion model and application to tomographic images. *Annals of Glaciology, 49*, 17-21. doi:10.3189/172756408787814834

**Author response to comments from referee #2 on tc-2021-53**

Referee comments are in black. Author responses are in blue and proposed changes in the manuscript are highlighted in **bold blue**.

**Comments from referee#2**

The authors presented a new dataset that combines BRF measurements and the X-ray tomography of the snow microstructure for two different snow morphological types. They found that faceted crystals exhibit a more anisotropic reflectance than fragmented particles, and the Malinka et al., (2016) model can generally reproduce the observed BRF using measured SSA. Different factors showed different importance for weak/intermediate and high absorption scenarios. The manuscript is generally well written, but there are still a few places that require further clarifications and explanations. Please see my comments below.

The authors thank the referee for these helpful comments. All comments have been accounted for and detailed responses are provided below.

Specific comments:
Section 2: Some descriptions of the measurement uncertainty and accuracy are needed, for example, for measurements of snow SSA and density as well as BRF.

We added uncertainty values for SSA and density. For BRF, the uncertainty is discussed P16 line 18-20. However we added more details in the methods section.

*List of modifications*

Page 4 line 23: "… with manual weighing. **The uncertainties on the SSA measured with DUFISSS and ASSSAP are in the range 10-15% for SSA smaller than 60 m2kg-1 (Gallet et al., 2009, Arnaud et al., 2011). For the density measured by manual weighing, the uncertainties are in the range 1 to 6 % (Proksch et al., 2016).** "
Page 5 line 7: "in a cold room at − 10°C. **The relative accuracy of the BRDF measurements is estimated of 1% in Bonnefoy et al., 2001. However, we don't believe that this accuracy is reached for high illumination angles (see Section 3.2)** "

**Added references:**
**Proksch, M., Rutter, N., Fierz, C., and Schneebeli, M.: Intercomparison of snow density measurements: bias, precision, and vertical resolution, The Cryosphere, 10, 371–384, https://doi.org/10.5194/tc-10-371-2016, 2016.**

Figure 1: It would be good to also provide microscopic images for the S2 and S3 samples (e.g., FC/DH) that are similar to Fig. 1a.

Figure 1 and its caption was modified as follows:

[Figure]

**Figure 1 : (A)-(C) Picture of snow from S1-S3 taken with a microscope and (D) experimental set-up for S2 sampling. The picture shows the inner part of the temperature gradient box (Calonne et al., 2014a) and the metallic sample holder for the optical measurements.**

Equation 3: What does the parameter "alpha" represent?

α is the argument in the characteristic function. Its physical meaning is discussed later, in section 3.1.2 (p 14 line 15): "if the argument α is equal to the substance absorption coefficient, then the characteristic function L(α) describes the process of photon absorption while travelling along random chords within an absorbing material, in this case, ice (see Malinka, 2014)".

To emphasize this, we inserted the argument explicitly:

"The mean chord length aj and the characteristic function of the ice phase **Lj(alpha) in** X-ray image *j* are obtained as"

Equation 4: How many images of a sample were used in the calculation in this study?

The number of images is varying with the samples (between 3 and 5). All the images used are described in Table 1.
This was added in the manuscript as follows:

Page 8 line 17 : "and N is the number of images of a sample (**see Table 1 for the number of images per sample**). "

Equation 5: It seems a little arbitrary to define x, y, z directions for a sample. How are these directions determined in this study? For example, did the authors assign the vertical direction of the snowpack layer as the z direction?

Z is the vertical direction of the snowpack layer, x and y are arbitrary in the plane perpendicular to z. z is a very important axis since it's aligned with gravity and with temperature gradient, which are then shaping the snow morphology.

We modified page 8 line 5 as follows: " by scanning the segmented images with "rays" along the (x,y,z)-directions. **The z-axis is aligned with the vertical direction of the snowpack while x, y are arbitrarily chosen in the plane perpendicular to the z-axis."**

Equation 7: Are the quantities (Sint, Vvp, Vint) all derived from the X-ray tomography images?
Yes, this has been added on page 9 line 14: "**where Sint, VVP and Vint are computed from the X-ray images.**"

Section 2.2.3: When computing the two SSA, did the authors used the image-averaged values for all the quantities in Equations 6 and 7?
In Figure 4, the two SSA are computed for each image separately.
The legend of Fig 4 has been modified accordingly and now reads: "respectively. **Each marker corresponds to one X-ray image.** Each sample …".
It is the same in Table 2 as indicated in the first column of the Table.

Section 2.5: The steps shown here use the Cst as another free variable in the fitting procedure. However, (1) the authors did not check if the retrieved Cst is reasonable (since the snow sample are new snow, I assume Cst should be very small). I saw that in Table 2, Cst is about 0.2-1 ppm, which is actually very large for soot content in snow and is typically for dirty snow samples.

Yes, Cst value for S1 is very high for alpine snow but it was taken in the urban area. The initial retrieval of Cst was done using the assumption of externally-mixed soot particles with the refractive index m = 1.75 + 0.44*i* at 550 nm (WMO, 1986; OPAC model). To be more consistent with recent studies, the retrieval of Cst has been redone using the recommendations of Tuzet et al. (2020): a mass absorption cross section of 11.25  $m^2kg^{-1}$ at 550 nm, the value recommended by Hadley and Kirchstetter (2012) and spectral dependence by the inverse wavelength $\lambda^{-1}$ (Bond and Bergstrom, 2006). This mass absorption coefficient value is an intermediate value between fresh BC and internally mixed aged BC (Tuzet et al., 2019). After the recalculation, the retrieved concentrations were about 500 ng/g for S1 and about 100 ng/g for S2 and S3.

*List of modifications*

Section 2.4 (Optical modelling) was changed as follows:

"**Further analysis showed that the albedo calculated with Eq. 10 was strongly overestimated in the green range of the spectrum. In particular, samples S2 and S3 with the geometrical thickness of 16.5 cm have the optical thickness   calculated with Eq. 10 of greater than 200. According to**

Malinka et al., 2016 such an optical thickness produces in the green range an albedo of the order of /( +4)~0.98, while the measured quantities reliably show a value of about 0.9. This means that the snow samples contain some light absorbing particles (Warren, 1982). These particles can be incorporated into the model. When their size is orders of magnitude smaller than that of snow grains (e.g. black carbon), scattering by these particles is negligible in comparison with scattering by snow grains. Thus, the effect of impurities can be modelled as an increase in the absorption coefficient of ice:

$$\alpha = \alpha_{ice} + \xi_{ice} * C_m, \quad (11)$$

where $\alpha$ is the resulting effective absorption coefficient of ice, $\alpha_{ice}$ is the absorption coefficient of pure ice, $\xi$ is the particle absorption cross section per its mass (mass absorption cross section), and $C_m$ is the relative mass concentration of absorbing particles.

The load and type of impurities were not measured in this experiment, so our choice of a pollutant was quite arbitrary. We assumed that the snow was polluted by black carbon with $\xi = 11.25$ m$^2$kg$^{-1}$ at 550 nm, the value recommended by Hadley and Kirchstetter (2012), and spectral dependence given the inverse wavelength $\lambda^{-1}$ (Bond and Bergstrom, 2006). The impurities internally-mixed in the ice phase lead to an increase in the mass absorption efficiency (Flanner et al., 2012 ; He et al., 2018). The value of 11.25 m$^2$kg$^{-1}$ is an intermediate value between fresh BC and internally mixed aged BC (Tuzet et al., 2019; 2020)."

Values in Table 2 of Cst (now Cm) have been updated due to the change of the mass absorption coefficient of the impurities.

Additional references:

WMO, 1986. A preliminary cloudless standard atmosphere for radiation computation. WCP 112, WMO/TD 24. https://library.wmo.int/index.php?lvl=notice_display&id=11668

A. Malinka, Stereological methods in the theory of light scattering by nonspherical particles, Springer Series in Light Scattering, Vol. 6, A. Kokhanovsky ed., Cham: Springer, 2021. — DOI:10.1007/978-3-030-71254-9_3

Tuzet, F., Dumont, M., Picard, G., Lamare, M., Voisin, D., Nabat, P., Lafaysse, M., Larue, F., Revuelto, J., and Arnaud, L.: Quantification of the radiative impact of light-absorbing particles during two contrasted snow seasons at Col du Lautaret (2058 m a.s.l., French Alps), The Cryosphere, 14, 4553–4579, https://doi.org/10.5194/tc-14-4553-2020, 2020.

Tuzet, F., Dumont, M., Arnaud, L., Voisin, D., Lamare, M., Larue, F., Revuelto, J., and Picard, G.: Influence of light-absorbing particles on snow spectral irradiance profiles, The Cryosphere, 13, 2169–2187, https://doi.org/10.5194/tc-13-2169-2019, 2019.

(2) Following (1), using Cst as a free variable in the fitting may bring uncertainty to SSA retrieval if the model-observation discrepancies that should have been attributed to SSA are attributed to Cst. Thus, using observed Cst or clean snow samples (Cst~0) to constrain the SSA retrieval or check if Cst is in a realistic range would help to improve the SSA retrieval accuracy.

The soot concentration affects albedo mostly in the visible range (e.g. He et al., 2018). The most sensitive range of spectrum to the mean chord $a$ (therefore, to SSA) is for wavelengths >0.9 um (e.g. Fig. 1 in Gallet et al., 2009) where the effect of soot is very small and even negligible. Consequently, Cst has very little influence on SSA retrieval.

**Additional reference :**
Gallet, J.-C., Domine, F., Zender, C. S., and Picard, G.: Measurement of the specific surface area of snow using infrared reflectance in an integrating sphere at 1310 and 1550 nm, The Cryosphere, 3, 167–182, https://doi.org/10.5194/tc-3-167-2009, 2009.

Figure 3: Does the retrieved albedo mean albedo calculated from the retrieved SSA? I did not see the description of albedo retrieval in Section 2.

The SSA is retrieved by fitting the simulated albedo with the measured one (see section 2.5 that gives details about the retrieval method). So yes, you can say that the retrieved albedo is the one calculated with the retrieved SSA.

To clarify this point, we modified the legend of Fig. 3, which now reads: "Measured and retrieved albedo spectra of samples S1 (left panel) and S2 (right panel). **The albedo is retrieved following the method described in section 2.5.**"

Figure 9: Would the systematic albedo overestimates at wavelengths < 1300nm also be due to the way of soot-snow mixing treated in the model? Recent studies have reported stronger albedo reduction by soot if soot is internally mixed with snow grains compared with soot-snow external mixing (e.g., He et al., 2018: https://doi.org/10.1002/2017JD027752; Flanner et al., 2012: https://doi.org/10.5194/acp-12-4699-2012). It is not clear how the soot-snow mixing is assumed in the optical modeling in this study based on Equation 11.

Concerning the soot-snow mixing, please see the response to your previous comment.
Note that Figure 9 is showing BRF and not albedo. For the albedo shown in Fig. 3, there is no systematic overestimation of the albedo by the model for wavelengths <1300 nm. The overestimation is observed for BRF at oblique incidence and viewing angles only (see Fig 13 first line). For example, at 700 nm, the model is overestimating the BRF only for viewing angles >=60°. This is also the case for S2 and S3 (see Figures B3 and B4). We attribute it to shortcomings of the model in the description of the angular dependence of the bidirectional reflectance, whether these shortcomings come from soot or not (e.g. https://tc.copernicus.org/articles/9/2323/2015/). This overestimation was also discussed at the end of p 23 and beginning of p 24, p 29 line 29-34 and p 30 in the first version of the manuscript.

*References*:
Peltoniemi, J. I., Gritsevich, M., Hakala, T., Dagsson-Waldhauserová, P., Arnalds, Ó., Anttila, K., Hannula, H.-R., Kivekäs, N., Lihavainen, H., Meinander, O., Svensson, J., Virkkula, A., and de Leeuw, G.: Soot on Snow experiment: bidirectional reflectance factor measurements of contaminated snow, The Cryosphere, 9, 2323–2337, https://doi.org/10.5194/tc-9-2323-2015, 2015.

The following modifications were done in the paper (page 16 line 26-27):
"**There is a little BRF overestimation in the visible and near IR range up to 1400 nm. Nevertheless, the albedo does not demonstrate any overestimation in this range of wavelengths. The BRF is overestimated mainly at oblique incidence and viewing angles and only in the visible range (see also Figs. 13, B3 and B4). We attribute the overestimation either to shortcomings of the measurements due to geometry (see above) or to the drawbacks of the model in the description of**

**the angular dependence of the bidirectional reflectance, which may not take into account several factors, such as dense packing or surface roughness.** ”

Page 23, Lines 13-19: It seems that the authors only described the model-observation differences in Figure 14 here without enough explanations on why the differences occur. It will be helpful if the authors could provide some insights into the causes.

On page 23, we initially wanted to only describe the figure but discussed the possible causes of model-measurements differences in Section 4 (Discussion and Conclusion), e.g. p 29 line 29-33 : “The anisotropy of the reflectance, quantified by the relative difference between the reflectances measured at 70° zenith angle in the forward and backward direction in the principal plane, is systematically overestimated in the simulations. This effect was reported by many authors and commonly attributed to the surface roughness (Warren et al., 1998; Painter and Dozier, 2004; Hudson et al., 2006; Jin et al., 2008; Carlsen et al., 2020), which was not accounted for in our simulations and generally leads to less anisotropic angular patterns.”

Page 23 line 13-19 were modified as follows:

“Figure 14 ….. The conclusions obtained at 1320 nm also stand at 800 nm (low absorption, Fig. B1). However at 1500 nm (high absorption, Fig. B2), the reflectance is underestimated almost twice for all angles and all samples. **The possible causes of this general overestimation of the anisotropy of the angular pattern are discussed in Section 4.**”

Similarly, more explanations of the model biases in Figure 15 will be helpful.
Please see response to your previous comment.
Page 24 line 4-8 is modified as follows:
“Figure 15 compares the anisotropy of the reflectance measured and simulated quantified by the anisotropy parameter η (Eq. 9). It shows that η is higher in the simulations than in the measurements, except for highly absorptive wavelengths (around 1500 and 2000 nm), where the differences between the samples are higher in the measurements than in the model. Also, the EXP and μCT simulations only differ for these highly absorptive wavelengths. **This general overestimation of the anisotropy in the simulations is also visible in Figs 9, 13, 14, B1-4. ”**

Figure 16: It seems that the uncertainty in ice refractive indices is not sufficient to explain the model-observation differences. Do the authors have any thoughts or speculations on other possible reasons that may contribute to this bias?

Yes, this was already discussed p30 lines 14- 23:
“Several hypotheses can be drawn to explain the model/measurements discrepancies. The first hypothesis is related to the imaginary part of the ice refractive index, to which the reflectance, including albedo, is very sensitive, but it is probably not sufficient to explain the different behaviours of the bidirectional reflectance distribution shown in Figs. 15 and 16. The second possible reason is the anisotropy of snow, which was noted by several authors (Calonne et al., 2012; Löwe et al., 2013) and for our samples ranged from 0.9 to 1.07, being calculated with Eq. (5). The last but most probable reason is the 'skin effect.' For small optical paths, the light penetration depth is really small and probably less than a mm (e.g. Mary et al., 2013), so the role of the skin layer, where the grains could be finer and have a preferable facet orientation, is crucial. This is also corroborated by the fact

that S2 and S3 have only a small difference in SSA, measured in deeper layers, but a strongly different behaviour in reflectance"

To improve the readability of the manuscript we modified the text as follows:

Page 25 line 1-5 : "For the other wavelengths, the model generally underestimates reflectance. The use of alternative ice refractive index values (Grundy and Schmitt, 1998) improves the simulations at 1420 and 1500 nm, **but is not sufficient to reconcile the measurements and the simulations for high viewing angles at 1500 and 2000 nm (see Sec. 4 for additional discussion)**. The difference between the µCT and EXP simulations are only noticeable for high viewing angles and high absorption. Under these conditions, accounting for the CLD shape retrieved from µCT leads to a decrease in the calculated reflectance in comparison with the simulations with exponential CLD. "

---

## Author Response (AR2)

Dear Editor,

Thanks for your review of the manuscript.
All technical corrections have been accounted for.
I am still waiting for the doi of the dataset (PANGAEA delay), I hope it will be ready after the typesetting.

Kind regards,

Marie Dumont on behalf of all co-authors.